# Acute exposure to apolipoprotein A1 inhibits macrophage chemotaxis in vitro and monocyte recruitment in vivo

**Asif J Iqbal[1†], Tessa J Barrett[2,3,4†], Lewis Taylor[1], Eileen McNeill[5,6,7], Arun Manmadhan[2,3,4], Carlota Recio[1], Alfredo Carmineri[1], Maximillian H Brodermann[1], Gemma E White[1], Dianne Cooper[8], Joseph A DiDonato[9], Maryam Zamanian-Daryoush[9], Stanley L Hazen[9], Keith M Channon[5,6,7], David R Greaves[1*‡], Edward A Fisher[2,3,4*‡]**

[1]Sir William Dunn School of Pathology, University of Oxford, Oxford, United Kingdom; [2]Division of Cardiology, NYU School of Medicine, New York, United States; [3]Department of Medicine, NYU School of Medicine, New York, United States; [4]Marc and Ruti Bell Program in Vascular Biology, NYU School of Medicine, New York, United States; [5]Division of Cardiovascular Medicine, University of Oxford, Oxford, United Kingdom; [6]John Radcliffe Hospital, Oxford, United Kingdom; [7]Wellcome Trust Centre for Human Genetics, University of Oxford, Oxford, United Kingdom; [8]William Harvey Research Institute, Queen Mary University of London, London, United Kingdom; [9]Department of Cellular and Molecular Medicine, Lerner Research Institute of the Cleveland Clinic, Cleavland, United States

**\*For correspondence:** david. greaves@path.ox.ac.uk (DRG); Edward.Fisher@nyumc.org (EAF)

†These authors contributed equally to this work
‡These authors also contributed equally to this work

**Competing interests:** The authors declare that no competing interests exist.

**Abstract** Apolipoprotein A1 (apoA1) is the major protein component of high-density lipoprotein (HDL) and has well documented anti-inflammatory properties. To better understand the cellular and molecular basis of the anti-inflammatory actions of apoA1, we explored the effect of acute human apoA1 exposure on the migratory capacity of monocyte-derived cells in vitro and in vivo. Acute (20–60 min) apoA1 treatment induced a substantial (50–90%) reduction in macrophage chemotaxis to a range of chemoattractants. This acute treatment was anti-inflammatory in vivo as shown by pre-treatment of monocytes prior to adoptive transfer into an on-going murine peritonitis model. We find that apoA1 rapidly disrupts membrane lipid rafts, and as a consequence, dampens the PI3K/Akt signalling pathway that coordinates reorganization of the actin cytoskeleton and cell migration. Our data strengthen the evidence base for therapeutic apoA1 infusions in situations where reduced monocyte recruitment to sites of inflammation could have beneficial outcomes.

## Introduction

CC chemokines are low-molecular-weight signalling proteins that mediate leukocyte trafficking in immune homeostasis and play a non-redundant role in monocyte/macrophage recruitment in pre-clinical models of inflammation (*Charo and Ransohoff, 2006*; *Mackay, 2008*; *White et al., 2013*). Macrophage chemotaxis towards CC chemokines and other pathophysiological chemoattractants, such as N-formylmethionyl-leucyl-phenylalanine (fMLF), complement component 5a (C5a) and chem-erin, are mediated by $G_{i/o}$ coupled G protein coupled receptors (*Allen et al., 2007*). Chemokine-mediated recruitment of monocytes and macrophages, in addition to retention and activation, are attractive areas for the development of novel anti-inflammatory agents that could find application in

**eLife digest** A molecule called cholesterol is an important component of the membranes found in cells and is also used to make some hormones and other useful molecules. However, cholesterol can also contribute to the formation of plaques in arteries, which can lead to a disease called atherosclerosis, the cause of heart attacks. Particles called high density lipoproteins (HDL) carry cholesterol around the body in the bloodstream and are thought to have anti-inflammatory properties. A protein called apoA1 is a major component of HDL particles and, acting as part of a HDL particle or alone, it removes cholesterol from cells.

Atherosclerotic plaques form when white blood cells collect in places where the arteries are inflamed. The membranes that surround the white blood cells contain receptors that are able to detect inflammatory signals called chemokines. These receptors eventually communicate with the machinery needed for cell movement. This machinery is concentrated in parts of the membrane known as lipid rafts. Iqbal, Barrett et al. investigated whether apoA1 can block the movement of mouse and human white blood cells towards the chemokines produced during inflammation.

The experiments show that apoA1 treatment strongly inhibited the movement of white blood cells towards a range of chemokines in a culture dish. The apoA1 protein removes cholesterol from lipid rafts in the membrane of the white blood cell, which changes the properties of the membrane and decreases the activity of the machinery needed for cell movement. Further experiments in mice with inflammation of the peritoneum, the thin layer of tissue that lines the inside of the abdomen, produced similar findings. The next step following on from this work would be to investigate whether apoA1 treatment can reduce the accumulation of white blood cells in mice that act as models of other inflammatory diseases, such as arthritis and atherosclerosis.

a wide range of chronic inflammatory pathologies (*Asquith et al., 2015*; *Schall and Proudfoot, 2011*; *Tabas and Glass, 2013*; *Zernecke and Weber, 2014*).

Apolipoprotein A1 (apoA1) is the major structural protein of high-density lipoprotein (HDL) and is widely recognized to have atheroprotective properties (*Fisher et al., 2012*). The cellular mechanisms that account for the observed anti-inflammatory effects of apoA1 and HDL, including those on innate immune cell activation, are incompletely understood. Postulated mechanisms include but are not limited to: (i) dependence on ATP-binding-cassette (ABC) A1, but independence of scavenger receptor class B type 1 (SR-B1) (*Murphy et al., 2008*); (ii) inhibition of NF-kB and phosphatidylinositol-3-kinase (PI3K) signalling (*Bursill et al., 2010*); and (iii) increased Activator of Transcription Factor 3 (ATF3) expression, which represses a subset of proinflammatory genes (*De Nardo et al., 2014*). Notably, the bulk of studies that have shown anti-inflammatory effects of apoA1 or HDL involve relatively long-term treatment of macrophages in culture or chronic exposure in vivo (*Diederich et al., 2001*; *Bursill et al., 2010*; *Ansell et al., 2003*). We thought it timely to focus on the acute effects of apoA1 treatment, in both in vitro and in vivo assays of monocyte/macrophage chemotaxis, a process of fundamental importance to the initiation and amplification of inflammation. We also explored the cellular mechanisms for anti-chemotactic effects and their dependence on cholesterol efflux.

## Results

### Acute exposure to apoA1 significantly reduces murine macrophage chemotactic responses

Chronic treatment of human monocytes with HDL and apoA1 has been reported to reduce chemotaxis to fMLF (*Diederich et al., 2001*). In the current study, we wanted to test whether acute exposure to the human form of apoA1 (referred to throughout as simply apoA1) could alter monocyte-derived cell chemotaxis to CC chemokines and in the process, to look for the earliest signalling events following addition of apoA1. To address this question, we used a real-time chemotaxis assay (*Iqbal et al., 2013*). Biogel elicited murine macrophages ($4 \times 10^5$/well) were pre-treated with apoA1 (40 µg/ml; a standard amount used in cholesterol efflux studies, e.g.*Adorni et al., 2007*) for 60 min at 37°C, 5% $CO_2$ and allowed to migrate towards 10 nM CCL2. The representative trace in

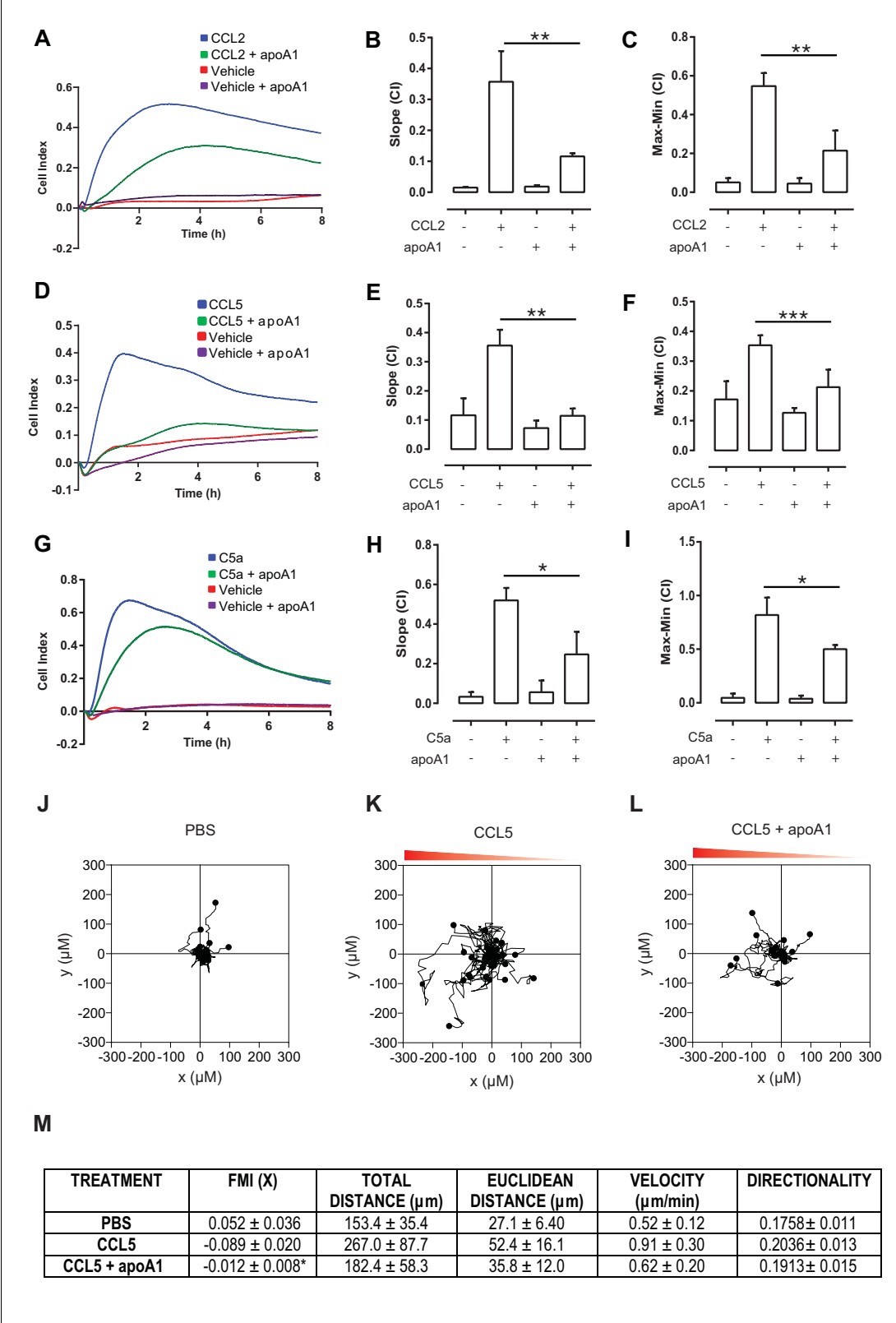

**Figure 1.** Acute pre-treatment with apoA1 inhibits macrophage chemotaxis. Biogel elicited macrophages were incubated with apoA1 (40 μg/ml) for 60 min before being added to the upper chamber (4 × 10⁵/well) of a CIM-16 plate and allowed to migrate for 7–8 hr toward 10 nM CCL2, CCL5, or C5a. Representative traces are shown in panels (**A, D, G**). Migration was measured with slope (**B, E, H**) and max-min analysis (**C, F, I**). Data expressed as mean + SEM, n = 4–8 biological replicates with 3–4 technical replicates per condition. Statistical analysis was conducted by one-way ANOVA with

*Figure 1 continued on next page*

*Figure 1 continued*

Dunnett's multiple comparison post-test. *p,0.05, **p,0.01, ***p,0.01 relative to CCL2, CCL5, or C5a alone. Biogel elicited macrophages from *Cd68*-GFP mice were incubated with either vehicle (**J** and **K**) or apoA1 (40 µg/ml) (**L**) for 60 min before being seeded into ibidiµ-Slide Chemotaxis³ᴰ slides (1.8 × 10⁴ cells/chamber). Migration of biogel elicited macrophages in the presence (**K** and **L**) or absence (**J**) of a CCL5 gradient (indicated by the red triangle) was followed by time-lapse microscopy and quantified by cell tracking. Data for FMI, Euclidean distance, total distance and velocity are summarised (**M**) and expressed as mean ± SEM of three independent experiments. Statistical analysis was conducted by an unpaired Students T test. *p,0.05, CCL5 vs CCL5 + apoA1.

The following figure supplement is available for figure 1:

**Figure supplement 1.** Short term incubations and different doses of apoA1 inhibit macrophage chemotaxis.

*Figure 1A* shows a clear reduction in chemotaxis towards CCL2 as quantified by slope and max-min analysis (67% reduction, $p \leq 0.01$, *Figure 1B–C*). A similar reduction in chemotaxis (71% reduction, $p \leq 0.01$) was observed in response to 10 nM CCL5 (*Figure 1D–F*). We also found that washing cells following acute exposure to apoA1 led to a similar impairment in migration as unwashed cells (data not shown). We extended our studies to show that this reduction in chemotaxis was seen with other, non-chemokine, chemoattractants: a 52% reduction in macrophage chemotaxis was observed in response to 10 nM C5a peptide ($p \leq 0.05$, *Figure 1G–I*), and a 80% reduction in chemotaxis to 10 nM chemerin, a potent chemoattractant of antigen presenting cells ($p \leq 0.01$, *Figure 1—figure supplement 1A–C*).

To confirm and extend our real-time chemotaxis observations, we performed an alternate macrophage migration assay that utilizes time-lapse microscopy to visualize individual cell movement in a chemoattractant gradient. Biogel elicited macrophages were treated for 60 min with apoA1 (40 µg/ml), or vehicle prior to plating in chamber slides and stimulating cells with 10 nM CCL5. Quantitative assessment of macrophage migratory behaviour was obtained by tracking individual cells using image analysis software (representative traces shown in *Figure 1J–L*). Analysis revealed that macrophage migration towards CCL5 was significantly impaired ($p \leq 0.05$) when cells were pre-treated with apoA1 as determined by the forward migration index (FMI) $-0.090$ v $-0.012$, (*Figure 1M*). Furthermore, treatment with apoA1 reduced the Euclidean distance and velocity of CCL5-stimulated macrophages to that of unstimulated controls, demonstrating a loss in cell directionality. Collectively, these findings support our observation that apoA1 directly impairs macrophage chemotaxis.

The initial time and concentration used for apoA1 treatment were based on several previous reports (*Murphy et al., 2011*, *2008*; *Diederich et al., 2001*). We wanted to test whether similar inhibitory effects on macrophage chemotaxis could be observed with shorter incubations and different doses of apoA1. Comparable to 60 min pre-treatment, both 20 min and 40 min were shown to significantly ($p \leq 0.001$) reduce macrophage chemotaxis to CCL5 by 75–86% (*Figure 1—figure supplement 1D–F*). Pre-treatment with 20 or 80 µg/ml of apoA1 for 60 min was also shown to reduce macrophage chemotaxis to CCL5 by 43% and 51% ($p \leq 0.05$), respectively (*Figure 1—figure supplement 1G–I*).

## Effects of apoA1 on cell migration require cholesterol efflux independent of ABCA1

ApoA1 functions in the first step of reverse cholesterol transport, cholesterol efflux from macrophages, through both ABCA1-dependent and -independent mechanisms (*Phillips, 2014*). Given the strong inhibitory effect of apoA1 in our chemotaxis studies, we were interested to see if the effects were mediated through cholesterol efflux, and if so, whether they were dependent on the cholesterol transporter, ABCA1. Our initial experiments used a recombinant apoA1 in which the Trp72 residue has been oxidized, thereby rendering apoA1 unable to efflux cholesterol via ABCA1-mediated processes (*Huang et al., 2014*). Notably, the migration capacity of macrophages to CCL5 treated with either recombinant wild-type apoA1 (rapoA1) or 5-OH-Trp72 rapoA1 (ox-rapoA1) was significantly inhibited (90–94% reduction, $p \leq 0.001$) (*Figure 2A–C*). We further demonstrated that the inhibitory effect of apoA1 to cell migration was independent of ABCA1-mediated cholesterol efflux by comparing the migratory ability of macrophages from C57BL/6 and *Abca1*⁻ᐟ⁻ mice. We found that macrophages from both C57BL/6 and *Abca1*⁻ᐟ⁻ mice have an equal capacity to migrate to CCL5

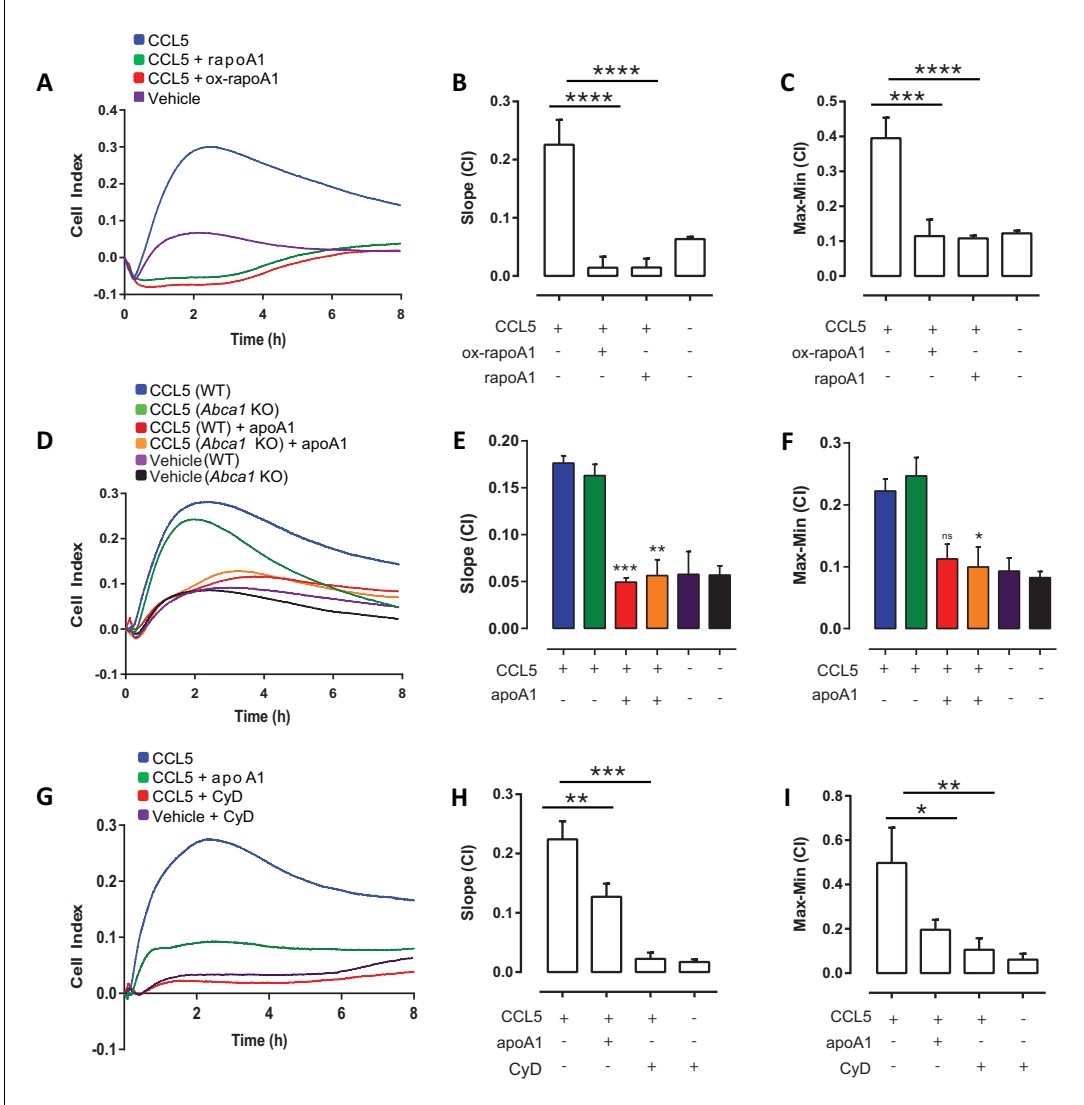

**Figure 2.** ApoA1 effects on macrophage chemotaxis are independent of ABCA1, but dependent on cholesterol efflux. Biogel elicited macrophages from WT, $Abca1^{-/-}$ mice were pre-incubated with apoA1, recombinant apoA1 (rapoA1) or 5-OH-Trp72-rapoA1 (ox-rapoA1) as described in Materials and methods before measuring chemotaxis towards 10 nM CCL5. Panels (A, D, G) show representative traces, migration was measured with slope (B, E, H) and max-min analysis (C, F, I). Biogel elicited macrophages from WT mice were pre-incubated with 3 mM methyl β-cyclodextrin (CyD) as described in Materials and methods before measuring chemotaxis towards 10 nM CCL5. Data are expressed as mean + SEM, n = 4–8 biological replicates with 3–4 technical replicates per condition. Statistical analysis was conducted by one-way or two-way ANOVA with Dunnett's multiple comparison post-test. *p,0.05, **p,0.01, ***p,0.001,****p,0.0001 relative to 10 nM CCL5 alone.

(*Figure 2D*), and that migration of both is inhibited by 62–70% (p≤0.05–0.001) following treatment with apoA1 for 60 min (*Figure 2D–F*). Treatment of macrophages with the non-specific cholesterol depleting agent methyl β-cyclodextrin (CyD) also led to markedly reduced macrophage chemotaxis to CCL5 (91% (p≤0.001); *Figure 2G–I*). Taken together, these results support a role for the contribution of a cholesterol-efflux pathway, independent of ABCA1, mediating the inhibitory effect of apoA1 on macrophage chemotaxis.

## Acute exposure to apoA1 significantly reduces human monocyte chemotaxis

Having shown significant effects of apoA1 on murine macrophage chemotaxis, we next tested the actions of apoA1 on human monocyte chemotaxis. Monocytes were isolated by CD14[+] selection

from human blood and pre-treated with apoA1 (40 µg/ml) or vehicle prior to assessment of their migratory capacity. The representative trace in *Figure 3A* shows a reduction in human monocyte chemotaxis to CCL2 following apoA1 treatment. Slope (*Figure 3B*) and max-min analysis (*Figure 3C*) of monocytes from multiple donors confirmed a significant 43% reduction in chemotaxis (p≤0.05) towards CCL2. A similar significant inhibition in monocyte migration (p≤0.05) was also observed in response to CCL5 (50% reduction) and C5a (42% reduction) (*Figure 3D–I*). A similar degree of inhibition was observed with the ibidi chamber assay, with the human monocyte migration towards CCL5 significantly impaired (p≤0.01) when cells were pre-treated with apoA1 (representative traces shown in *Figure 3J–L*) as determined by the forward migration index (FMI) −0.246 v 0.064, (*Figure 3M*). Furthermore, treatment with apoA1 significantly reduced the Euclidean distance, total distance, velocity and directionality of CCL5-stimulated human monocytes.

Previous studies have reported an inhibitory role for apoA1 on monocyte adhesion to endothelial cells (*Murphy et al., 2008*). We therefore used an in vitro flow chamber assay to assess the effects of apoA1 on peripheral blood mononuclear cell (PBMC):HUVEC interactions under flow. Pre-incubation of PBMCs with apoA1 considerably reduced cell capture as shown in the representative images (*Figure 3—figure supplement 1A*). PBMC rolling, adhesion and transmigration were all significantly reduced by 20–45% following pre-treatment with apoA1 for 45 min (*Figure 3—figure supplement 1B–D*). Integrins play a key role in the process of leukocyte recruitment (*Campbell and Humphries, 2011*), so we measured levels of active CD11b/ total CD11b, CD49b, CD62L and C5aR1 on human monocytes that had been treated for 45 min with apoA1 and then stimulated with C5a for 5 min. No differences were observed in the expression of any of these markers (*Figure 3—figure supplement 1E–J*) indicating that the effects of acute apoA1 treatment were independent of receptor and adhesion molecule modulation.

## Acute apoA1 pre-treatment of monocytes ex vivo impairs recruitment to sites of inflammation following adoptive transfer

To determine whether the inhibitory actions observed with acute apoA1 treatment on macrophage and human monocyte chemotaxis in vitro extends to monocyte/macrophage trafficking in vivo, we performed adoptive transfer studies with GFP[+] monocytes. We previously reported that monocytes from *Cd68*-GFP mice express high levels of GFP transgene and that the level of expression increases as they differentiate into macrophages, greatly facilitating their detection and recovery (*Iqbal et al., 2014*). *Figure 4A* summarizes the experimental design used to assess the effect of apoA1 treatment ex vivo on monocyte recruitment in vivo. Briefly, bone marrow monocytes were isolated by negative immunomagnetic selection (typical purity of 65–70% GFP[+] monocytes (*Figure 4B*)) and then treated with either apoA1 (40 µg/ml) or vehicle for 45 min. Monocytes were then washed and injected (i.v) into C57BL/6J recipient mice that had previously received 100 µg zymosan (i.p) to initiate a mild inflammatory response (*Iqbal et al., 2014*).

Recruitment of GFP[+] monocytes to the peritoneal cavity was assessed by flow cytometry, with representative plots shown in *Figure 4C*. Acute pre-treatment with apoA1 ex vivo reduced GFP[+]-monocyte recruitment to the peritoneal cavity of recipient mice by 65% (p≤0.01) when compared to vehicle treated cells (*Figure 4E*). This occurred despite similar levels of GFP[+] monocytes in the blood of mice that received adoptively transferred apoA1 or vehicle treated monocytes (*Figure 4G*). Importantly, total monocytes recruited to the peritoneal cavity and circulating in the blood were equivalent at 16 hr in all groups of recipient mice following zymosan challenge (*Figure 4D and F*), highlighting that the effects of treatment were restricted to only the monocytes pre-treated with apoA1. In a parallel set of experiments, we used SJL/1 mice (which express CD45.1) as bone marrow monocyte donors and C57BL/6J mice (which express CD45.2) as recipients to confirm results obtained with *Cd68*-GFP monocytes, and observed a similar defect in recruitment following acute pre-treatment with apoA1 (*Figure 4—figure supplement 1 A–E*).

## Mice expressing the human apoA1 transgene have impaired leukocyte recruitment during acute inflammation

All experiments conducted thus far have focused on the effects of acute apoA1 treatment. We wanted to also investigate what effect chronic exposure to apoA1 may have on leukocyte recruitment in vivo. To address this we used mice carrying the human apoA1 transgene (*Apoa1*Tg) under

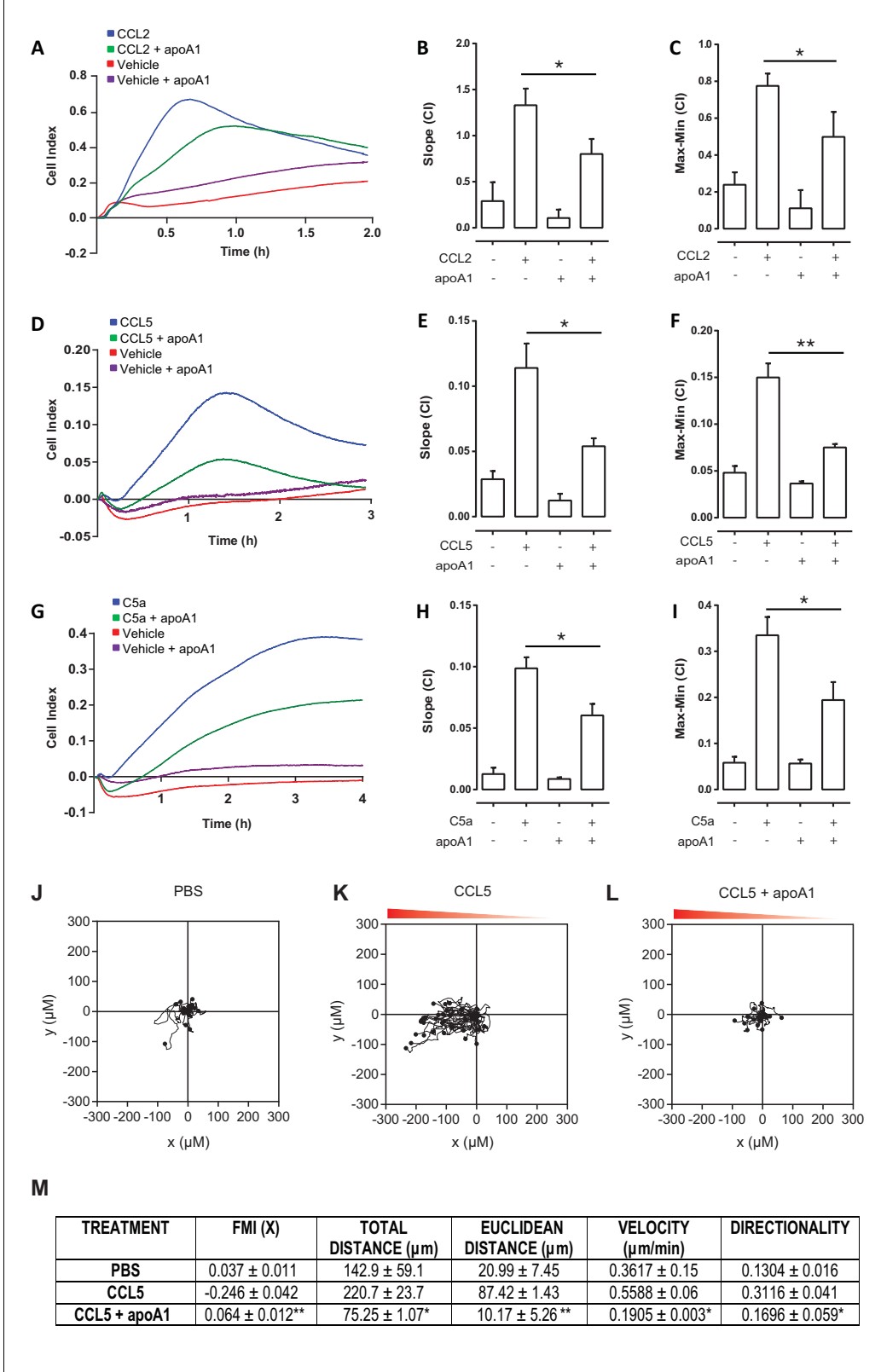

**Figure 3.** Acute pre-treatment with apoA1 reduces human monocyte chemotaxis. CD14[+] selected human monocytes ($2 \times 10^5$) were pre-treated with apoA1 (40 µg/ml) for 60 min before being added to the upper chamber of a CIM-16 plate and allowed to migrate for 2 hr toward 10 nM CCL2, CCL5 or C5a. Panels (**A**, **D**, **G**) show representative traces, and migration was measured with slope (**B**, **E**, **H**) and max-min analysis (**C**, **F**, **I**). Data are expressed as mean + SEM, n = 4 biological replicates with 3–4 technical replicates per condition. Statistical analysis was conducted with one-way ANOVA with

*Figure 3 continued on next page*

*Figure 3 continued*

Dunnett's multiple comparison post-test. *p,0.05, relative to chemoattractant alone. CD14[+] selected human monocytes were pre-treated (J and K) with apoA1 (40 µg/ml) (L) for 60 min before being seeded into ibidi µ-Slide Chemotaxis[3D] slides (1.8 × 10[4] cells/chamber). Migration of human monocytes in the presence (K and L) or absence (J) of a CCL5 (50 nM) gradient (indicated by the red triangle) was followed by time-lapse microscopy and quantified by cell tracking. Data for FMI, Euclidean distance, total distance, velocity and directness are summarised (M) and expressed as mean ± SEM of two independent experiments. Statistical analysis was conducted by an unpaired Students T test. *p,0.05, **p,0.05, CCL5 vs CCL5 + apoa1.

The following figure supplement is available for figure 3:

**Figure supplement 1.** Acute pre-treatment with apoA1 reduces PBMC:HUVEC interactions under flow.

the control of its natural promoter, which results in elevated circulating levels of total apoA1, with >90% being human apoA1 (*Rubin et al., 1991*; *Chajek-Shaul et al., 1991*). *Apoa1*Tg and C57BL/6J control animals were injected (i.p) with 100 µg of zymosan to provide an inflammatory stimulus (*Iqbal et al., 2014*). Peritoneal exudates 16 hr post-challenge contained predominately neutrophils and monocytes as identified as CD45[+]/CD115[-]/Ly6G[+] and CD45[+]/CD115[+]/Ly6GC[hi/lo] populations by flow cytometry, respectively (representative plots shown in *Figure 5A*). Total cell recruitment in *Apoa1*Tg mice was significantly reduced by 27% (p≤0.05) when compared to C57BL/6 controls, with a reduction in total neutrophils (26%, p≤0.05) and monocytes (30%, p≤0.01) (*Figure 5B–D*). Elevated apoA1 levels significantly suppressed the recruitment of the Ly6C[hi] monocyte subset to peritoneal cavity (42%, p≤0.05, *Figure 5E*). No changes in Ly6C[lo] subset or in total blood white cell counts were observed between any of the experimental groups (*Figure 5F–G*). Chemokines CCL2 and CXCL1 were measured in the peritoneal exudates of mice from the 4 hr and 16 hr time-points, but no significant differences were observed (CXCL1 was non detectable at 16 hr) (*Figure 5H–J*). Similar levels of chemokines, but reduced neutrophil and monocyte influx, suggests that changes in the intrinsic properties of the leukocytes were responsible for the reduction in innate immune cell recruitment.

## ApoA1 impairs macrophage activation of Akt following CCL5 stimulation

Given the consistent effects of apoA1 on monocyte and macrophage chemotaxis in vitro and in vivo, we were interested to identify the mechanistic basis of these findings. We hypothesized that there was an interruption or inhibition of an aspect of chemokine signalling that regulates cell migration. In our initial studies, we found no differences in G-protein coupled receptor 'upstream' phenomena, such as cAMP levels, $Ca^{2+}$ mobilisation or β-arrestin recruitment as a consequence of apoA1 pre-treatment (*Figure 6—figure supplement 1*). Hence, we turned our attention to other known downstream signalling events that are essential for cytoskeletal reorganization, including PI3K and Rho/Rac (*Vorotnikov, 2011*). Human monocytes were pre-treated with apoA1 (40 µg/ml) for 60 min, prior to C5a stimulation. Pre-treatment with apoA1 reduced Akt phosphorylation by 54%, as determined by western blotting and densitometry (*Figure 6A*). A similar significant reduction in Akt signalling was also observed following apoA1 and CyD pre-treatment of biogel elicited murine macrophages (reduction of 32% and 63% respectively, *Figure 6B–C*). In addition to Akt phosphorylation, we also measured the effects of apoA1 on its upstream activator, PI3K. Pre-treatment of RAW macrophages with apoA1 for 1 hr prior to stimulation with C5a led to a reduction in PI3K activity by 36% (*Figure 6D*), and PI3K inhibitor studies revealed significant reduction of macrophage chemotaxis in vitro (*Figure 6E–F*). Taken together, these results provide evidence that apoA1 modulates PI3K/Akt signalling resulting in a significant impairment of monocyte and macrophage chemokine-induced migration.

## Decreased Akt cell signalling by apoA1 is associated with modulation of membrane lipid rafts

Given that signalling in the PI3K/Akt pathway regulating cell movement involves proteins and lipids associated with membrane lipid rafts (*Zajchowski and Robbins, 2002*), we hypothesized that the apoA1-mediated suppression of Akt phosphorylation in monocytes and macrophages occurred due to perturbation of lipid raft content as a consequence of cholesterol efflux. To assess this, biogel-

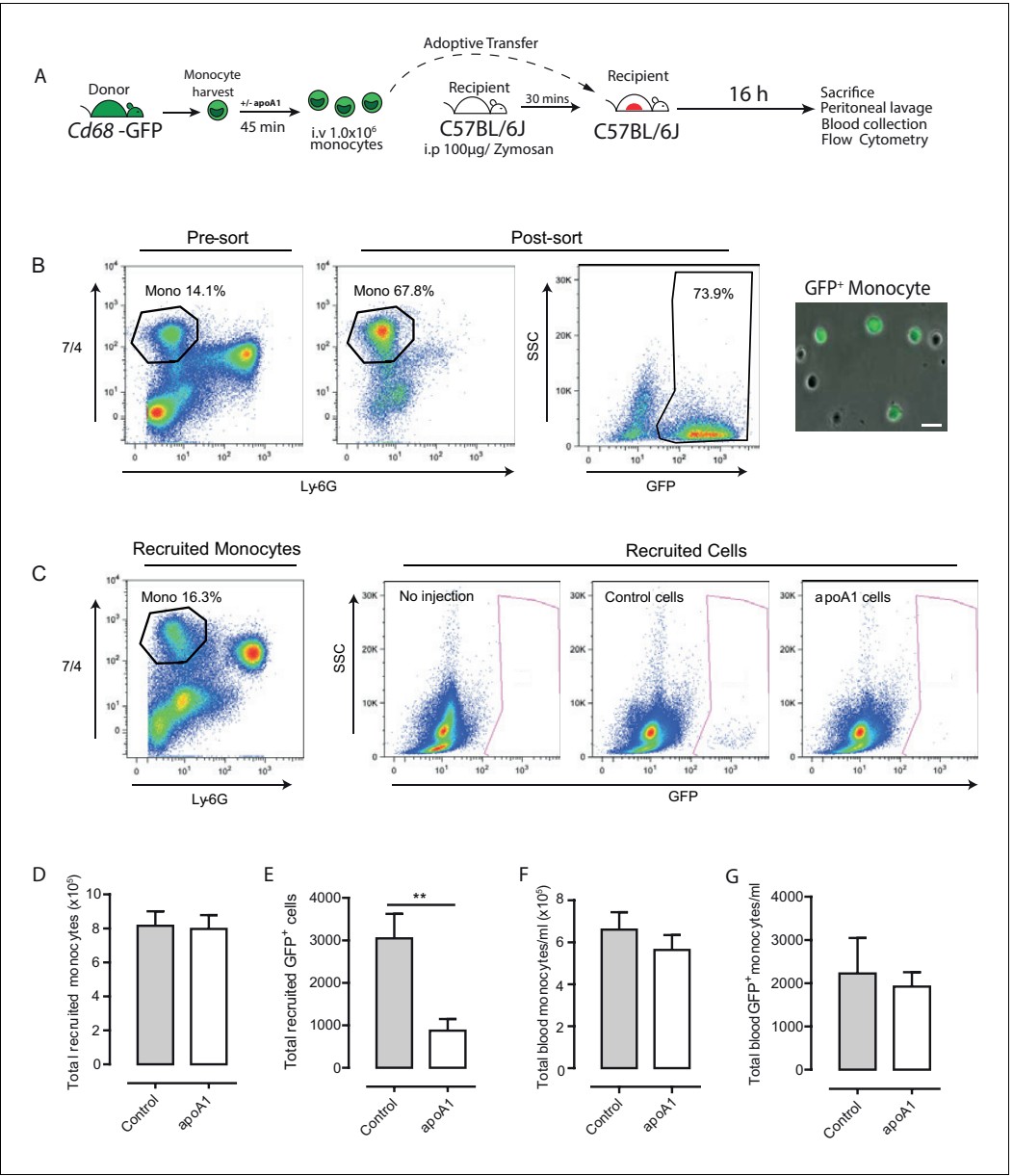

**Figure 4.** Acute exposure to apoA1 reduces the recruitment of adoptively transferred monocytes to sites of inflammation in vivo. Diagram of the experimental design presented in (**A**). Monocytes were isolated from *Cd68*-GFP bone marrow using 'no-touch' negative immunomagnetic selection (**B**). Isolated monocytes were characterized as 7/4$^{high}$/Ly6G$^{low}$, with a typical yield of 65–70% monocytes, of which 70–80% were GFP positive. Isolated *Cd68*-GFP monocytes (1 × 10$^6$) treated for 45 min with either apoA1 (40 μg/ml) or PBS and were adoptively transferred into C57BL/6J mice by i.v. injection 30 min after i.p. injection with 100 μg zymosan. Mice were euthanized at 16 hr, and peritoneal lavage and blood samples were collected. Representative flow cytometry plots of peritoneal lavage from C57BL/6J mice that received adoptively transferred GFP positive monocytes treated with or without apoA1 during ongoing zymosan-induced peritonitis (**C**). Total monocytes and total adoptively transferred GFP$^+$ monocytes were quantified in peritoneal lavage (**D–E**) and blood (**F–G**). Data are expressed as mean + SEM of 8–10 mice from two independent experiments. Statistical analysis was conducted by paired Students T test. **p,0.01, relative to control.

The following figure supplement is available for figure 4:

**Figure supplement 1.** Acute exposure to apoA1 reduces the recruitment of adoptively transferred CD45.1 monocytes to sites of inflammation in CD45.2 recipient mice.

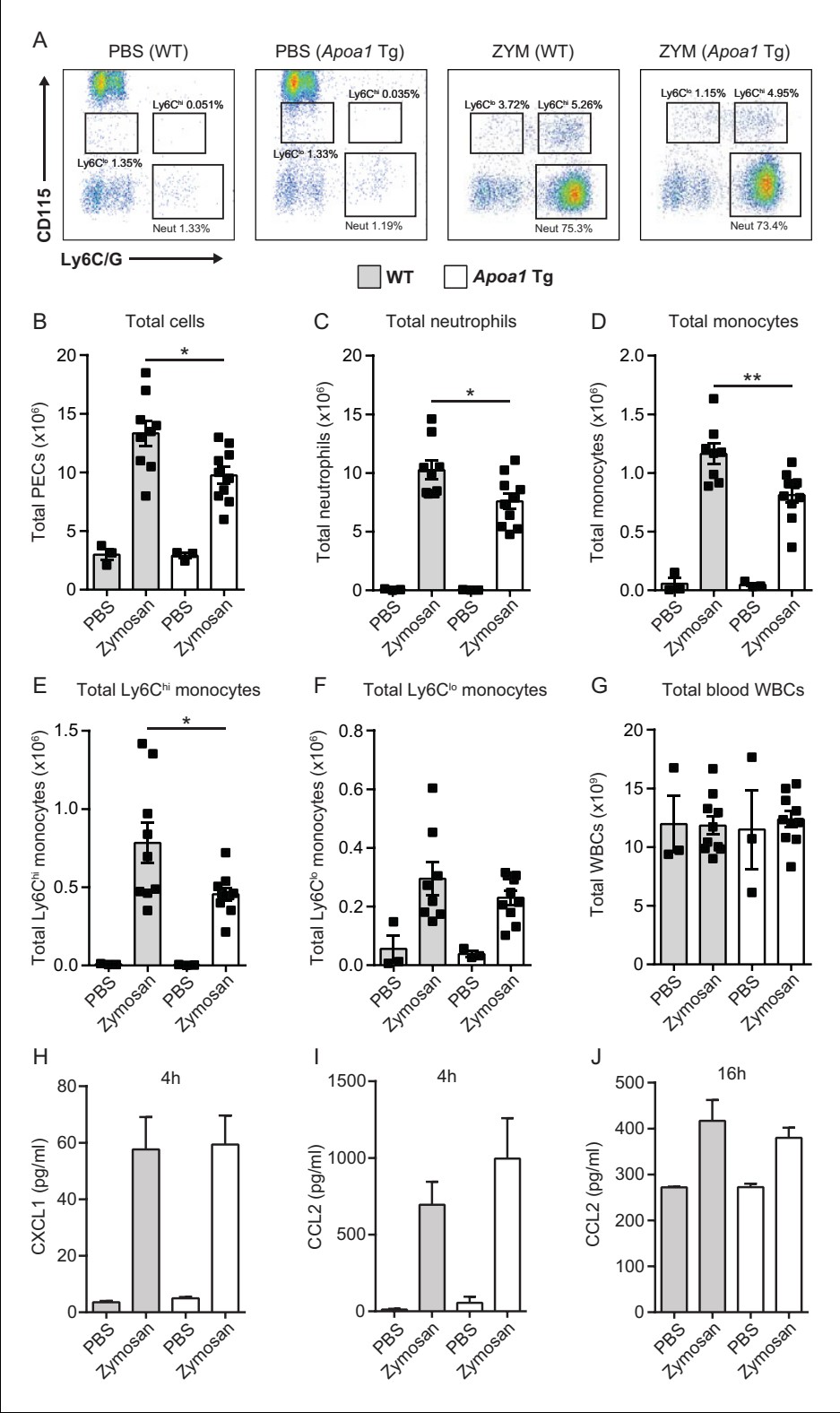

**Figure 5.** *Apoa1*Tg mice display reduced leukocyte recruitment in zymosan peritonitis. *Apoa1*Tg and WT control mice were injected i.p. with 100 μg zymosan. Mice were euthanized at 16 hr and peritoneal lavage samples were collected. Representative flow cytometry plots are shown in panel (**A**) of neutrophils (Neut) characterised as CD45+/CD115-/Ly6G+ and monocytes characterised as CD45+/CD115+/Ly6Chi/lo from peritoneal lavage fluid. Total cells (**B**), neutrophils (**C**) total monocytes (**D**), Ly6Chi monocytes (**E**), Ly6Clo monocytes (**F**) within the peritoneal

*Figure 5 continued on next page*

*Figure 5 continued*

cavity following zymosan challenge are expressed. (G) Total circulating blood white cell counts from *Apoa1*Tg mice and WT controls are expressed as mean ± SEM of 3–9 mice per condition. (H–J) CXCL1 and CCL2 levels were measured in peritoneal lavage fluid by ELISA. Data are expressed as mean + SEM of 2–6 mice per genotype of two independent experiments. Statistical analysis was conducted by one-way ANOVA with Dunnett's multiple comparison post-test. *p,0.05, **p,0.01 compared to WT.

elicited macrophages (*Figure 7A–F*) or human THP-1 macrophages (*Figure 7G–J*) were treated with CyD or apoA1 or its variants, and the abundance of the raft-enriched components caveolin-1 and ganglioside $G_{M1}$ assessed. Pre-treatment with apoA1, rapoA1, ox-rapoA1, or CyD led to a significant reduction in total cell caveolin-1 (*Figure 7A–B*) as determined by western blotting, and ganglioside $G_{M1}$ expression as determined by flow cytometry (*Figure 7C*) or confocal fluorescence microscopy (*Figure 7D–F*). Membrane lipid rafts are enriched in cholesterol, and perturbations to this component leads to raft reorganization, which affects the activity of receptors and raft-bound signalling factors (*Simons and Toomre, 2000*). To determine whether apoA1 reorganized cholesterol-rich domains, sucrose density gradient ultracentrifugation of human macrophages was performed and raft markers identified (*Figure 7G*). As previously reported (*Adorni et al., 2007*; *Koseki et al., 2007*), cellular free cholesterol is distributed between both low (rafts) and high (non-rafts) density fractions (*Figure 7H*). Treatment with apoA1 (40 µg/ml), ox-rapoA1 (40 µg/ml), or CyD leads to a significant reduction in cholesterol content in both raft and non-raft fractions, demonstrating that apoA1 can significantly modulate the composition of this signalling nanodomain (*Figure 7I–J*).

## Discussion

Diseases in which chronic inflammation is a central feature of their pathology are significant contributors to mortality and serious morbidities worldwide. A major factor that sustains inflammation in these diseases is a mobilization and activation of the innate immune system (*Tabas and Glass, 2013*). Chronic inflammation is characterized by the continued recruitment of circulating monocytes, in part mediated by their response to chemokines, such as CCL2 and CCL5, which are synthesized locally in response to tissue damage and resultant locally-generated inflammatory mediators (*Tabas and Glass, 2013*). After entering the tissue, recruited monocytes become macrophages and the degree of their activation is frequently judged in vitro by the chemotactic response to chemokines. Thus, the identification and development of agents with the ability to inhibit monocyte/macrophage chemotaxis to various chemokines is an extremely active area of investigation with the ultimate goal of deploying such agents as therapeutics to combat chronic inflammatory diseases, including arthritis and atherosclerosis (*Allen et al., 2007*; *Koelink et al., 2012*; *Koenen and Weber, 2011*).

It has been previously reported that both HDL and its major protein component apoA1 can inhibit leukocyte chemotaxis to various chemokines (*Bursill et al., 2010*; *Wang et al., 2010*; *Ansell et al., 2003*). Previous studies have been heavily weighted towards experimental conditions in which the assays were exclusively performed in vitro or the duration of treatment was prolonged ($\geq$24 hr). Furthermore, the mechanistic basis for the effects on chemotaxis has been largely unexplored. In this present report, we have addressed these issues in both murine and human monocytes and macrophages in vitro and in models of inflammation in vivo. Our major findings are that: (1) only a short exposure (<1 hr) of monocytes or macrophages to apoA1 is required to reduce chemotaxis to chemoattractants in vitro and most notably, recruitment to an inflammatory site in vivo; (2) apoA1 treatment reduces both the velocity and the directionality of macrophage migration to an established chemokine gradient; (3) inhibition of monocyte-derived cell chemotaxis requires cholesterol depletion of the cells, through a mechanism independent of ABCA1, a cholesterol transporter previously invoked to explain certain anti-inflammatory effects of apoA1 (*Murphy et al., 2008*), but rather through cholesterol efflux most likely by aqueous diffusion (*Adorni et al., 2007*). We propose a model in which apoA1 and CyD treatment disrupts plasma membrane lipid rafts by cholesterol depletion, which, in turn, dampens the signalling pathway required for reorganization of the actin

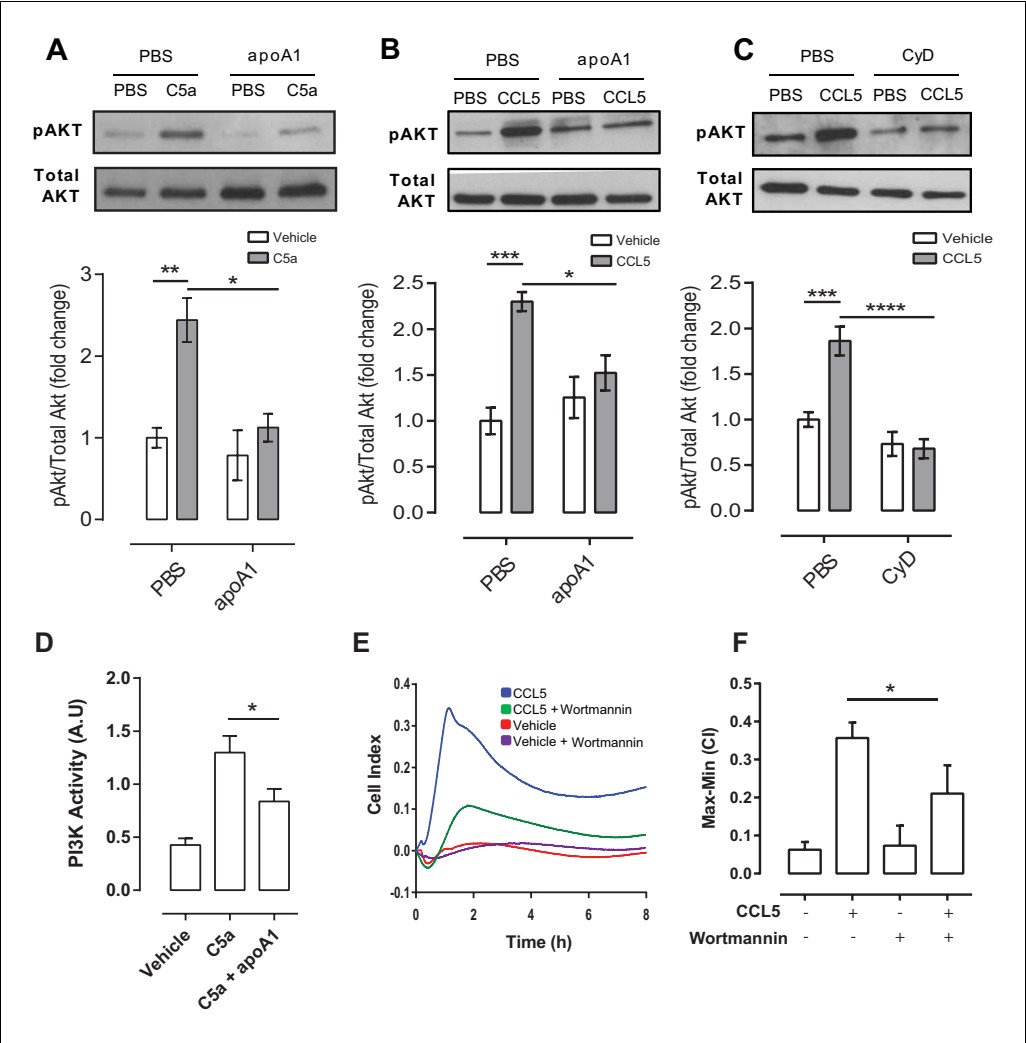

**Figure 6.** ApoA1 suppresses Akt signalling and PI3K activity following chemokine activation of human and murine monocyte-derived cells. CD14[+] selected human monocytes or biogel elicited murine macrophages ($2 \times 10^5$) were pre-incubated with either vehicle (PBS), apoA1 (40 µg/ml) or 3 mM methyl β-cyclodextrin (CyD) for 60 min as described in Materials and methods. Human CD14[+] monocytes (**A**) or murine macrophages (**B**,**C**) were then stimulated with either 10 nM C5a or CCL5. Relative levels of phosphorylated Akt were determined by Western blotting. Representative blots from independent biological replicates are shown. Densitometry of western blots is shown in panels below representative western blots (n > 2/group). Data are expressed as mean ± SEM, n = 2–6 biological replicates. Murine RAW macrophages were pre-incubated with apoA1 (40 µg/ml) or vehicle for 60 min at 37°C, 5% $CO_2$ prior to stimulation with C5a (10 nM) for 10 min (**D**). PI3K was immunoprecipitated from cell lysates and PI3K activity determined by incubating with $PIP_2$ and measuring $PIP_3$ production by competitive ELISA. Data are expressed as mean + SD, n = 4 independent experiments. Statistical analysis was conducted by one-way ANOVA with Dunnett's multiple comparison post-test. *p<0.05, relative to C5a alone. Biogel elicited macrophages were incubated with 100 nM wortmannin for 60 min before being added to the upper chamber ($4 \times 10^5$/well) of a CIM-16 plate and allowed to migrate for 8 hr at 37°C, 5% $CO_2$ towards 10 nM CCL5. Representative traces are shown in panel (**E**). Migration was measured by max-min analysis (**F**). Statistical analysis was conducted by one-way ANOVA with Dunnett's multiple comparison post-test. *p<0.05, ***p<0.001 relative to CCL5 alone.

The following figure supplement is available for figure 6:

**Figure supplement 1.** $G\alpha_{i/o}$-signalling, β-arrestin recruitment and intracellular $Ca^{2+}$ flux remain unaffected in response to apoA1.

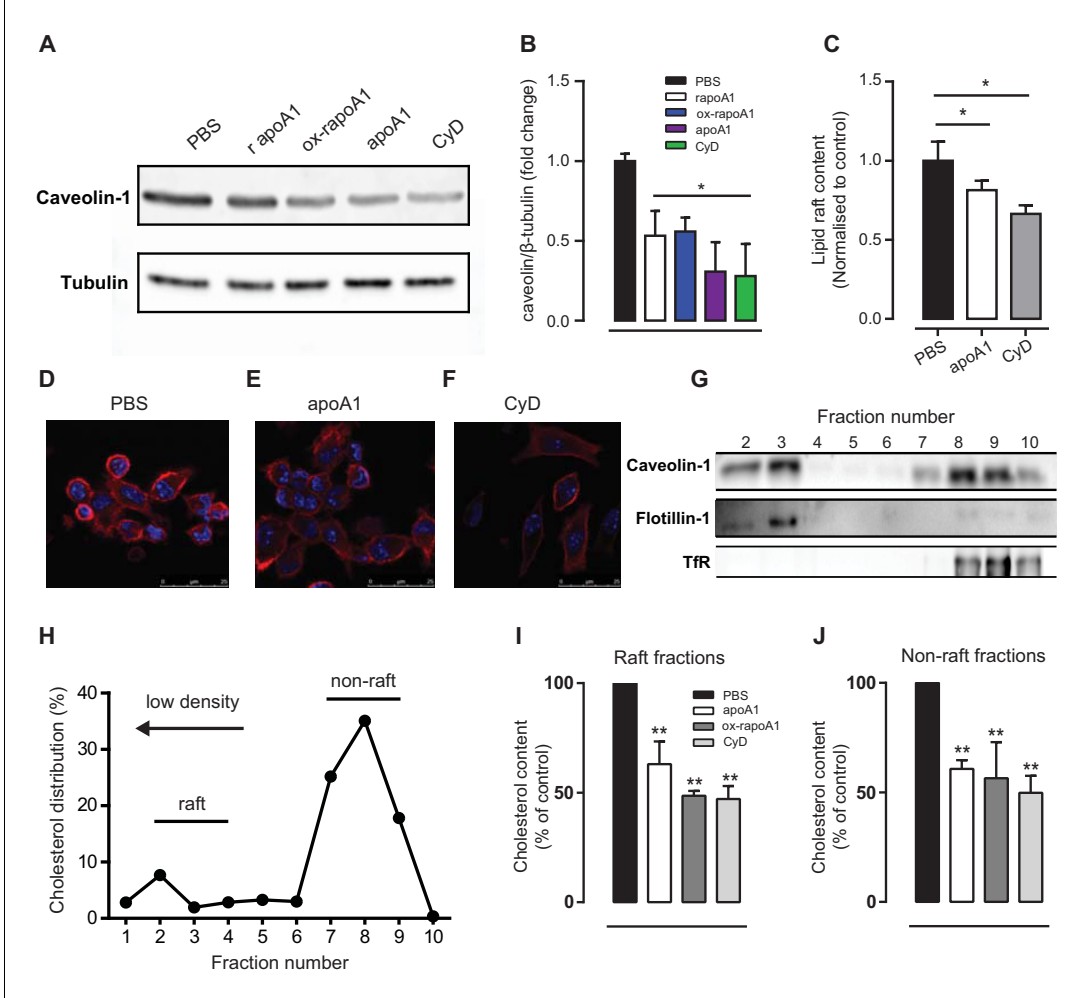

**Figure 7.** ApoA1 modulates monocyte-derived cell lipid raft cholesterol content in cells of murine and human origin. (A–B) Total cell caveolin-1 was assessed by Western blotting following treatment of biogel elicited macrophages with apoA1 (40 µg/ml), ox-rapoA1 (40 µg/ml) or 3 mM methyl β-cyclodextrin (CyD) for 1 hr. The lipid raft content of biogel elicited macrophages was assessed following treatment with apoA1 (40 µg/ml) or 3 mM CyD for 1 hr, and incubation of the cells with cholera toxin B Alexa Fluor 647 conjugate to bind lipid raft ganglioside $G_{M1}$. Relative fluorescence (compared to PBS-treated control cells) was assessed by flow cytometry (C), and cells were imaged by confocal microscopy (D–F). Prior to fractionation of membrane lipid rafts, human THP-1 macrophages were treated with either vehicle, apoA1 (40 µg/ml), ox-rapoA1 (40 µg/ml), or 3 mM CyD for 1 hr prior to stimulation with 10 nM CCL5 for 5 min. Cells were then homogenized in a 0.2% Triton-X 100 containing buffer prior to 5–45% sucrose density gradient centrifugation. Fractions were collected from the top (fraction 1, lightest fraction) to the bottom (fraction 10, heaviest fraction), and fraction proteins separated by SDS-PAGE, transferred to PVDF membranes and caveolin-1, flotillin-1, or transferrin receptor (TfR) content assessed by western blotting. Representative distribution of raft and non-raft marker proteins presented in panel (G). The cholesterol content of each fraction was determined via the amplex red assay, with a representative distribution shown in panel (H). The sums of the cholesterol content in lipid raft fractions (fractions 2–4), and non-raft fractions (fractions 8–10) following treatment with apoA1, ox-rapoA1 and CyD are shown in panels (I) and (J), respectively. Western blots are representative of three independent experiments. Cholesterol contents are means + SEM of three independent experiments. Statistical analysis was conducted by one-way ANOVA with Dunnett's multiple comparison post-test. *p<0.05, **p<0.01 relative to PBS.

The following figure supplement is available for figure 7:

**Figure supplement 1.** ApoA1 depletes membrane lipid rafts of cholesterol in cholesterol-loaded THP-1 cells.

cytoskeleton, as reflected by impaired PI3K/Akt activation. These findings will now be discussed in more detail.

ApoA1 or HDL treatments of monocytes/macrophages in vitro and of mice have typically been of longer duration than used in this current report, sometimes reaching one week (*Smythies et al.,*

*2010*; *Hewing et al., 2014*). This makes it difficult to discern early direct effects of apoA1 from those that are downstream and likely to include many subsequent indirect effects. For example, in Bursill *et al.*, a 24 hr treatment of monocytes with recombinant HDL resulted in reduced chemotaxis, and this was associated with decreased expression of CCR2 and CX3CR1 (*Bursill et al., 2010*). These long-term effects are unlikely to be the basis for the observed ~80% reduction in macrophage chemotaxis following only 20 min of exposure to apoA1 (*Figure 1—figure supplement 1D–F*). In our experiments, important "upstream" or "receptor proximal" signalling functions of chemokine G protein–coupled receptors (GPCRs) were not affected by acute treatment with apoA1 i.e. decreased intracellular cAMP levels (*Figure 6—figure supplement 1A*), β-arrestin recruitment (*Figure 6—figure supplement 1B*) and $Ca^{2+}$ flux (*Figure 6—figure supplement 1C–D*). In the current studies, both murine (*Figure 1A*), and human monocyte migration to CCL2 (*Figure 3A*) were inhibited by apoA1 treatment. Importantly, in some chemotaxis experiments, apoA1 was removed after pre-treatment of cells, yet with even short pre-treatment times (as little as 20 min) an ~80% inhibition of chemotaxis to CCL5 was still observed (*Figure 1—figure supplement 1D–F*). In other words, continuous exposure to apoA1 was not required for significant inhibition of chemotaxis.

Our in vivo studies recapitulated important features of assays performed on immune cells in vitro. For example, continuous and prolonged exposure of leukocytes to apoA1 and HDL, inherent to the *Apoa1*Tg mouse model, was associated with reduced recruitment of innate immune cells to sites of inflammation (*Figure 5*). As with the experiments in vitro which involve long-term incubation of cells with apoA1 or HDL, early vs. late effects cannot be discerned using this transgenic mouse model. Thus, it was especially notable that in our adoptive transfer approach, only 45 min ex-vivo exposure of monocytes to apoA1 was required to reduce their recruitment to an inflammatory site by 65% (*Figure 4* and *Figure 4—figure supplement 1*).

That the rapidity of these effects in vitro and in vivo parallels the acute loss of lipid rafts in monocytes (*Murphy et al., 2008*) or macrophages (*Figure 7*) is not likely to be a coincidence. Rather, this rapid desensitization to the chemoattractant effects of chemokines likely represents the perturbation of lipid rafts, which are crucial to the common signalling machinery downstream from chemokine receptors that are required for the reorganization of the cytoskeleton and integral to chemotaxis (*Vorotnikov, 2011*). Consistent with this is our finding that incubation with apoA1 was effective in blunting chemotaxis to multiple chemokines and chemoattractants, yet the early events following receptor ligation were unimpaired (*Figure 6—figure supplement 1*). More directly, we show using independent assays (western blotting and flow cytometry) that lipid raft abundance was reduced after incubation of macrophages with apoA1, as was Akt phosphorylation. Akt activation in response to phosphorylation by PI3K is required for chemotaxis and was confirmed by our experiments using the PI3K inhibitor, wortmannin (*Figure 6E–F*).

Given the importance of lipid raft organization to chemotaxis, we were interested in the mechanism by which apoA1 mediated its effects. Lipid rafts are enriched in cholesterol and its depletion is known to change the content and function of many of its components (*Simons and Toomre, 2000*). While it is long been appreciated that apoA1 is a major acceptor of cellular cholesterol, and therefore can affect the lipid raft structure and functions, most studies of apoA1's anti-inflammatory effects have focused on cholesterol efflux mediated by specific membrane factors, in particular ABCA1 (e.g. *Wang et al., 2001*; *Murphy et al., 2008*; *Pagler et al., 2011*). By multiple criteria, our results are consistent with a non-specific cholesterol depletion process (sometimes called aqueous diffusion (*Adorni et al., 2007*) that was responsible for the impairment in chemotaxis, as: (1) similar effects of apoA1 incubation were observed with *Abca1*$^{-/-}$ cells (*Figure 2D–F*); (2) CyD mediated cholesterol efflux was equally effective as apoA1 (*Figure 2J–L*); and (3) a mutant form of apoA1 (oxrapoA1, or 5-OH-Trp72 apoA1), which is a poor acceptor of cholesterol via ABCA1, also inhibited chemotaxis (*Figure 2A–C*). It should be noted that under the conditions of our assays using non-cholesterol loaded cells, ~80% of cholesterol efflux is likely through the aqueous diffusion mechanism (*Adorni et al., 2007*), and this dominance is consistent with our results. Furthermore, that apoA1 can promote cholesterol efflux in *Abca1*$^{-/-}$ macrophages has also been reported (*Zhu et al., 2008*). In addition, the inhibitory effect of apoA1 on macrophage chemotaxis to CCL5 is not affected by cholesterol loading of the cell (data not shown), consistent with apoA1's ability to promote cholesterol removal from membrane lipid rafts of these cells (*Figure 7—figure supplement 1*).

Continued monocyte recruitment and macrophage activation are hallmarks of sites of chronic inflammation. CC chemokines play a key role in monocyte recruitment and macrophage activation in

pre-clinical models of human diseases characterised by chronic inflammation (*Charo and Ransohoff, 2006*; *White et al., 2013*; *Tabas and Glass, 2013*). Extending the findings from chemokine receptor knockout mice to interventional studies in man using chemokine receptor antagonists has proven challenging with multiple drugs that target single chemokine receptors failing to progress beyond early phase randomised clinical trials (*Schall and Proudfoot, 2011*; *Koelink et al., 2012*). As an alternative to targeting single macrophage chemoattractant GPCRs for therapeutic benefit, we and others have explored the potential of targeting multiple chemokines using chemokine binding proteins (*White et al., 2011*) or blocking macrophage responses to multiple chemoattractants using netrin (*van Gils et al., 2012*) or apoA1 (this study). The effects of netrin on macrophage chemotaxis are dependent on the Unc5b receptor while our experiments show that the effects of apoA1 occur via a receptor-independent mechanism by changing lipid raft organisation leading to decreased PI3K activity and hence decreased monocyte/macrophage adhesion, rolling and chemotaxis.

The present results are relevant to our recent studies of apoA1's anti-tumorigenic activity (*Zamanian-Daryoush et al., 2013*). This effect was attributed, in part, to suppression of the recruitment and accumulation of myeloid-derived suppressor cells (MDSC), potent stimulators of tumor growth. Notably, MDSC recruitment to tumors was significantly reduced in both apoA1 Tg mice and in apoA1-deficient mice receiving apoA1 therapy. MDSCs are comprised of a mixture of myeloid cells with granulocytic and monocytic morphology. The ability of apoA1 to suppress the recruitment of MDSCs to tumors, therefore, is in agreement with our current results showing the ability of apoA1 to suppress myeloid cell migration to inflammatory sites. While the exact molecular mechanism behind apoA1's anti-tumor effect is currently unknown, given the similarities between the results in the cancer and inflammation settings, it is likely that apoA1's ability to modulate membrane lipid raft cholesterol content influences signalling pathways regulating MDSC recruitment to tumors.

Using recombinant apoA1 particles, Nissen and colleagues showed regression of coronary atherosclerosis in patients given 5 weekly intravenous injections (*Nissen et al., 2003*; *Kingwell et al., 2014*), and it is tempting to speculate that the efficacy was due, at least in part, by the mechanisms illustrated by our studies. Our in vivo findings (*Figures 4*, *5* and *Figure 4—figure supplement 1*) suggest that apoA1 infusion may also be therapeutic in other clinical situations where reduced monocyte recruitment to sites of inflammation would be beneficial. In future studies, it will be interesting to test the effect of apoA1 or CyD infusion in pre-clinical models of ischemia reperfusion injury (mimicking a myocardial infarction), septic shock or ischemic stroke, all of which are situations where limiting the initial rush of inflammatory leukocytes into sites of injury could have a significant effect on subsequent tissue injury and repair processes.

## Materials and methods

### Materials

All cell culture media and buffers were obtained from PAA systems (Yeovil, UK) or Gibco (Life Technologies, NY) unless otherwise specified. Laboratory chemicals were obtained from Sigma-Aldrich (Gilligham, Dorset, UK or St. Louis, MO). Chemokines and other chemoattractants were purchased from Peprotech (London, UK or Rocky Hill, NJ), Merck Millipore (Feltham, UK or Billerica, MA) and R&D Systems (Abingdon, Oxford, UK or Minneapolis, MN). Inhibitors were purchased from TOCRIS (Bristol, UK). Human apolipoproteinA1 (apoA1) was purchased from Merck Millipore and Academy Bio-medical Co. (Houston, Texas). Recombinant apoA1 (rapoA1) was prepared as described previously using a genetically engineered *E. coli* mutant strain deficient in endotoxin synthesis (*Huang et al., 2014*) and 5-hydroxy Trp72-apoA1 (ox-rapoA1) was prepared using an orthogonal tRNA synthetase pair that incorporates 5-hydroxy tryptophan (*Ellefson et al., 2014*).

### Animals

UK animal studies were conducted with ethical approval from the Dunn School of Pathology Local Ethical Review Committee and in accordance with the UK Home Office regulations (Guidance on the Operation of Animals, Scientific Procedures Act, 1986). Male (10–14 weeks) C57BL/6J and SJL/1 mice (CD45.1) mice were obtained from Harlan Laboratories (Oxfordshire, UK). Female *Cd68*-GFP mice (6–8 weeks) were obtained from our in-house colony. All USA animal experiments were carried out according to the guidelines of the National Institutes of Health and approved by the New York

University Institutional Animal Care and Use Committee (Protocol 102090). Male (10–14 weeks) C57BL/6J, and C57BL/6-Tg(APOA1)1Rub/J (*Apoa1*Tg) mice were obtained from the Jackson Laboratory, Maine, USA. *Abca1*$^{-/-}$ mice were kindly provided by Kathryn J Moore (NYU School of Medicine, USA).

### Human monocyte isolation

Human blood from anonymous healthy donors was obtained in the form of leukocyte cones from the NHS Blood and Transplant service. Leukocyte cones contain waste leukocytes isolated from individuals donating platelets via apharesis, and consist of a small volume (~10 ml) of packed leukocytes with few red blood cells or platelets. For monocyte isolation, blood was diluted with 1:3 with PBS followed by separation using ficoll gradient centrifugation as previously described (*White et al., 2014*). Following human peripheral blood mononuclear cell (PBMC) isolation and washing, a total of $1.25 \times 10^8$ PBMCs were labelled and positively selected with CD14 micro-beads and MACS separation (Miltenyi Biotec, Bisley, Surrey, UK). Monocytes were resuspended at $4 \times 10^6$ cells/ml in chemotaxis buffer (RPMI 1640/25 mM HEPES/0.5% (w/v) BSA) and left on ice until required.

### Biogel elicitation of primary mouse macrophages

Mice were injected i.p. with 1 ml of sterile 2% biogel P-100 fine polyacrylamide beads (45–90 μm; Bio-Rad Laboratories, Hertfordshire, UK or Hercules, CA) suspended in PBS. Mice were sacrificed 4 days later and the peritoneum lavaged with 10 ml of ice-cold PBS/2 mM EDTA.

### Adoptive transfer of *Cd68*-GFP bone marrow monocytes into mice with ongoing zymosan-induced peritonitis

Monocytes were isolated from *Cd68*-GFP bone marrow using EasySep Mouse Monocyte Enrichment kit (Stem Cell Technologies, France) as described previously (*Iqbal et al., 2014*). Briefly, bone marrow cell suspensions were passed through a 70 μm cell strainer and red blood cells lyzed (Pharmlyse, BD Bioscience, SJ) according to the manufacturer's instructions. The bone marrow cell suspension was treated with the EasySep reagents and monocytes isolated by depletion with an EasyPlate magnet (Stem Cell Technologies) using a 96 well plate for 5 min, followed by a second selection for a further 2 min. The purity of the resulting populations confirmed by flow cytometry using antibodies specific for CD45 (BD Bioscience), CD11b (BD, Bioscience), 7/4 (AbD Serotec, Oxford, UK) and Ly-6G (Biolegend, London, UK). Bone marrow isolations from a total of 4 femurs were typically found to yield $1–2 \times 10^6$ cells at a purity of 65–85% monocytes. Monocytes were incubated with apoA1 (40 μg/ml) for 45 min at 37°C in 5% $CO_2$ and washed twice before i.v. administration. C57BL/6J mice were administered 100 μg zymosan A in PBS (Sigma-Aldrich) i.p. 30 min before i.v. administration of $1 \times 10^6$ isolated monocytes (70–80% GFP positive, in 200 μl). After 16 hr mice were sacrificed and peritoneal exudates were collected by lavage with 5 ml of ice cold sterile PBS/2 mM EDTA. Total cell counts and cellular composition of peritoneal exudate were determined as previously described (*Iqbal et al., 2014*).

### ACEA xCELLigence real-time cell migration

Experiments were carried out with CIM-16 plates and an xCELLigence RTCA-DP instrument (ACEA, San Diego, USA) as previously described (*Iqbal et al., 2013*). Chemoattractants were made to desired concentrations and loaded into the lower wells of the CIM-16 plate. Upper wells were filled with chemotaxis buffer and plates equilibrated for 30 min at RT. Biogel elicited macrophages were resuspended in chemotaxis buffer and incubated with apoA1 or other treatments for 20–60 min at 37°C, 5% $CO_2$. Cell suspensions were placed into the wells of the upper chamber, and the assay performed over 8 hr (5 s data points).

### Murine monocyte-derived macrophages

Bone marrow monocyte derived macrophages (BMDM) were generated as previously described (*Iqbal et al., 2013*; *Marim et al., 2010*). Briefly, fresh bone marrow cells from tibiae and femurs of C57BL6/J mice (8–12 weeks) were flushed and cultured in DMEM supplemented with 10% heat-inactivated fetal bovine serum (FBS), 2 mM l-glutamine, 100 U/ml penicillin and 15% supernatant from L929 cells as a source of macrophage colony stimulating factor (*Englen et al., 1995*). BMDM were

generated by culturing $4 \times 10^6$ bone marrow cells in 10 ml of medium in 100 mm non-tissue culture treated petri dishes (Sterilin, Abergoed UK). On day 3, an additional 5 ml of medium was added. Cells were harvested with PBS/2 mM EDTA on day 7.

## Zymosan-induced peritonitis model

C57BL/6J or apoA1 TG mice were administered 100 µg zymosan A (Sigma-Aldrich) in PBS. After 16 hr mice were sacrificed and peritoneal exudates collected by lavage with 5 ml of ice cold sterile PBS with 2 mM EDTA. Total cell counts and cellular composition of peritoneal exudate were determined as previously described (*Iqbal et al., 2014*). Antibodies used to identify neutrophils (CD45[+]/CD115[lo]/Ly6G[+]) and monocytes (CD45[+]/CD115[hi]/Ly6C[hi/lo]) were APC anti-mouse Ly-6G/Ly-6C (Gr-1), PE anti-mouse CD115 (CSF-1R), and PE/Cy7 anti-mouse CD45 from Biolegend (San Diego, CA). Total blood cell counts were determined with a Genesis blood counter (Oxford Science, CT).

## ibidi 3D chemotaxis

Biogel elicited macrophages were filtered through a 40 microns cell strainer, to remove the biogel beads, centrifuged at 300 g for 5 min and then resuspended in chemotaxis buffer at a concentration of $3 \times 10^6$ cells/ml. Filtered biogel elicited macrophages were then incubated with either vehicle or apoA1 (40 µg/ml) for 45 min at 37°C, 5% $CO_2$, before being seeded into ibidi Chemotaxis 3D µ–slides. Cells were then allowed to adhere for 30 min before chemoattractant or the vehicle was added to the appropriate side of the chemotaxis plate (as per the manufacturer's instruction). Cells were then imaged every 5 min for 60 frames using an Olympus FV1200 inverted confocal microscope equipped with a heated and humidified live cell imaging stage. Macrophages (20–30 per chamber) were then tracked using the manual tracking plugin within ImageJ software and chemotaxis analysis conducted using the chemotaxis and migration tool software (*Zengel et al., 2011*).

## Flow chamber assay

Slides VI 0.4 (IB, Germany) were coated with 0.5% bovine gelatin and seeded with human umbilical vein endothelial cells (HUVEC; passage 2–3) such that monolayers were confluent the next day. HUVEC were stimulated with TNF-a (10 ng/ml, 7 hr; Sigma-Aldrich) prior to commencing the flow assay. PBMCs were freshly isolated as previously described (*Cooper et al., 2008*) from healthy volunteers using a double density gradient of histopaque (Sigma-Aldrich). Briefly, blood was diluted 1:2 with RPMI, layered over histopaque and centrifuged at 400 ×g for 30 min. The PBMC layer was removed and washed twice. PBMCs were pre-incubated with apoA1 for 45 min at 37°C, diluted to $1 \times 10^6$ cells/ml in PBS-BSA (0.1%) and perfused over HUVEC at a flow rate of 1 dyne/cm$^2$ for 5 min. Six random fields/treatment were recorded for 10 s each. The total number of interacting PBMCs was quantified as captured and further classified as rolling (cells that moved in the direction of flow over the 10 s analysis period), adherent (cells that remained stationary over the 10 s analysis period) or transmigrated (cells that migrated through the endothelial monolayer) using Image-Pro Plus software (MediaCybernetics, Buckinghamshire, UK).

## Human monocyte flow cytometry

PBMCs were freshly isolated as previously described (*Cooper et al., 2008*) from healthy volunteers using a double density gradient of histopaque (Sigma-Aldrich). PBMCs were pre-incubated with apoA1 for 45 min at 37°C, diluted to $1 \times 10^6$ cells/ml in PBS-BSA (5%) and stimulated with C5a (10 nM) for 5 min and then stained with active CD11b (Clone CBRM 1/5; eBioscience), total CD11b (Clone ICRF44; Biolegend), CD49b (Clone P1E6-C5; Biolegend), CD14 (Clone 63D3; Biolegend), C5aR1 (Clone S5/1; Biolegend) and CD62L (Clone DREG-56; Biolegend) and analysed by flow cytometry.

## Lipid raft isolation and characterization

Human THP-1 monocytes were cultured in RPMI containing 10% (v/v) FBS at 37°C in 5% $CO_2$. For differentiation to macrophages, THP-1 monocytes were seeded at a density of $1.2 \times 10^6$ cells/ml (in 150 cm$^2$ dish) and incubated with phorbol 12-myristate 13-acetate ((PMA) 100 ng/mL) for 72 hr in RPMI with 10% FBS. Prior to treatments, differentiated macrophages were serum starved in RPMI containing 0.2% BSA overnight. THP-1 macrophages were then incubated with apoA1 (40 ug/ml),

methyl-β-cyclodextrin (3 mM) or vehicle (PBS) for 1 hr prior to stimulation with 10 nM C5a for 5 min. For cholesterol loading experiments differentiated macrophages were incubated with methyl-β-cyclodextrin:cholesterol complexes (0.5 mg/mL) for 1 hr, prior to apoA1 incubation. Lipid rafts were prepared according to methods described previously (*Airiian and Mkrtchian, 1987*; *Gaus et al., 2005*). Following treatment, cells were washed three times with ice-cold PBS, centrifuged (100 ×g, 10 min, 4°C) and resuspended in 1 ml of ice-cold lysis buffer (25 mM MES, pH 6.5, 150 mM NaCl, 0.2% Triton X-100, 1 mM PMSF, 1 mM NaF, 0.1 mM $Na_3VO_4$, 5 µg/ml leupeptin, 5 µg/ml aprotinin). Cells were then sonicated three times on ice for 15 s, and lysates centrifuged (100 ×g, 10 min, 4°C). The supernatant was gently mixed with an equal volume of 90% (w/v) sucrose in MBS (25 mM MES, pH 6.5, 150 mM NaCl). Two millilitres of the mixture was overlaid with 2 ml of 35, 30, 25, and 5% (w/v) sucrose (all in MBS). The sucrose gradient was then spun at 23,000 ×g at 4°C in a Beckman SW41 rotor for 16–20 hr. Ten fractions of 1 ml were collected from the top and analysed for protein, and cholesterol. Free cholesterol content of each fraction was measured using the Amplex Red cholesterol assay (Invitrogen), and protein content by the Pierce 660 nm protein assay. To assess the distribution of raft and non-raft marker proteins, fractions were mixed with 4x Laemmli buffer (250 mM Tris-HCl, pH 6.8, 8% SDS, 40% glycerol, 0.004% bromophenol blue, 20% β-mercaptoethanol), heated at 95°C for 5 min, resolved by SDS-PAGE gels, and transferred to PVDF membranes. Membranes were blocked with 5% BSA in TBS-T for 1 hr and then incubated with either rabbit anti-caveolin (1:2000) (BD Biosciences), mouse anti-flotillin1 (1:1000), mouse anti-TfR (1:1000) diluted in 5% BSA/TBS-T overnight at 4°C. Membranes were then incubated with an HRP-conjugated anti-rabbit or anti-mouse secondary antibodies (1:20,000) for 1 hr at RT. Protein bands were visualised by incubating the membranes for 5 min with Amersham ECL prime and chemiluminescent detection.

In some experiments, lipid rafts were characterized in biogel elicited macrophages ($1 \times 10^6$) after incubation with PBS, apoA1 (40 µg/ml) or methyl-β-cyclodextrin (CyD, 3 mM) for 1 hr prior to staining membrane rafts with cholera toxin subunit B conjugated with Alexa Fluor 647 (Life Technologies) for 5 min at 4°C. Cells were then fixed and membrane lipid raft fluorescence determined by flow cytometry, or visualized by microscopy.

## Intracellular cAMP measurement

Intracellular cAMP levels were measured using Discoverx cAMP Hunter eXpress kits (DiscoveRx, Birmingham, UK) following the manufacturer's protocol. Briefly, CHO-K1 cells overexpressing human C5aR1 receptor were plated into a ½ area 96 well plate (15,000 cells/well) and incubated at 37°C, 5% $CO_2$ for 24 hr. The medium was then removed and replaced with assay buffer containing a cAMP capture antibody. Cells were pre-treated for 1 hr at 37°C, 5% $CO_2$ with either vehicle or 40 µg/ml apoA1 prior to being stimulated with either vehicle or C5a at the indicated concentration for 30 min at 37°C, 5% $CO_2$. Cell lysis and cAMP detection were then performed as per the manufacturer's protocol. Luminescence measurements were taken using a PHERAstar microplate reader (BMG Labtech, Aylesbury, UK).

## β-arrestin recruitment assay

Recruitment of β-Arrestin was measured using the DiscoveRx PathHunter eXpress β-Arrestin GPCR Assay following the manufacturer's protocol. Briefly, cells were seeded into ½ area 96 well plates and incubated at 37°C, 5% $CO_2$ for 48 hr prior to testing. Agonist or vehicle was then added to the corresponding wells and the plate incubated at 37°C, 5% $CO_2$ for 90 min. Cell lysis and detection reagents were subsequently added and one hour later, luminescence measurements were taken using a PHERAstar microplate reader.

## Intracellular Ca²⁺ measurements

Biogel elicited macrophages were seeded into black walled 96 well microplates and left for 4 hr at 37°C, 5% $CO_2$. The medium was then removed and replaced with RPMI 1640 containing 4 µM FURA2-AM and 0.04% pluronic acid (Life Technologies) supplemented with either vehicle or 40 µg/ml apoA1. Plates were left for 45 min at RT in the dark. Subsequently, cells were washed twice with PBS, the medium replaced with Fluorobrite DMEM (Life technologies) and the plate placed into a PHERAstar microplate reader set to 37°C. Macrophages were then stimulated with either vehicle

or C5a at the indicated concentration and changes in FURA-2 fluorescence were measured using excitation wavelengths of 340 nm and 380 nm and an emission wavelength of 520 nm.

## Protein assay and western blots

Biogel elicited macrophages ($2 \times 10^6$) were pre-incubated with PBS, apoA1 or 3 mM methyl-β-cyclo-dextrin (Sigma-Aldrich) for 1 hr and then stimulated with 10 nM CCL5 for 5 min, washed and lysed in ice cold lysis buffer (150 mM NaCl, 0.8 mM $MgCl_2$, 5 mM EGTA, 50 mM HEPES, 1% IGEPAL CA-630) supplemented with protease and phosphatase inhibitors (Sigma-Aldrich). Protein concentration was determined using a BCA protein assay kit (Thermo Fisher Scientific, Loughborough, UK) following the manufacturer's protocol. Samples were then diluted 3:1 with 4x Laemmli buffer (250 mM Tris-HCl, pH 6.8, 8% SDS, 40% glycerol, 0.004% bromophenol blue, 20% β-mercaptoethanol) and heated at 95°C for 5 min. Samples (30 μg of protein) were resolved on SDS-PAGE gels and transferred onto Hybond ECL nitrocellulose membranes (GE Healthcare, Buckinghamshire, UK). Membranes were blocked with 5% BSA in TBS-T for 2 hr at RT or overnight at 4°C. After blocking, membranes were incubated with rabbit anti-phospho-Akt (1:2000) (Cell Signalling Technology, MA), or rabbit anti-β-tubulin (1:2000) (EMD Millipore) diluted in 5% BSA/TBS-T for 2 hr at RT or overnight at 4°C. Membranes were then incubated with an HRP-conjugated anti-rabbit secondary antibody (1:20,000) for 1 hr at RT. Protein bands were visualised by incubating the membranes for 5 min with Amersham ECL prime and subsequent exposure to x-ray film over a range of exposure times. To confirm equal protein loading between samples, bound antibodies were removed by incubating the nitrocellulose membranes in stripping buffer (60 mM Tris-HCl pH 6.8, 2% SDS, 0.8% β-mercaptoe-thanol) for 30 min at 50°C. Membranes were blocked with 5% BSA in TBS-T for 2 hr at RT and then incubated with rabbit anti-Akt diluted in 5% BSA/TBS-T for 2 hr at RT. Protein band detection was conducted as described above. Densitometry was performed with Li-cor Image Studio Lite 4.0 (Li-Cor, Cambridge, UK)

## PI3K activity measured by ELISA

RAW 264.7 cells were incubated with PBS, apoA1 (40 μg/ml) or methyl-β-cyclodextrin (3 mM) for 1 hr prior to stimulation with C5a (10 nM). Following treatments, cells were washed three times with 137 mM NaCl, 20 mM Tris HCl pH7.4, 1 mM $CaCl_2$, 1 mM $MgCl_2$, 0.1 mM $Na_3VO_4$ (buffer A), and then lysed in 1 ml of buffer A supplemented with 1 mM phenylmethylsulphonyl fluoride, and 1% NP-40. Lysed cells were then centrifuged, and the supernatant incubated with 5 μl anti-PI3K p85 anti-body (ABS233, N-SH2 domain, 0.36 mg/mL) (Millipore) and incubated overnight at 4°C. To immu-noprecipitate PI3K, 60 μl of 50% slurry of Protein A-agarose beads was added, and samples incubated for 2–4 hr at 4°C. Immunoprecipitated enzyme was collected by centrifugation at 9300 ×g for 5 s. Pellets were washed three times in buffer A plus 1% NP-40, three times in 0.1 M Tris-HCl, pH 7.4, 5 mM LiCl, 0.1 mM $Na_3VO_4$ and twice with 10 mM Tris-HCl, pH 7.4, 150 mM NaCl, 5 mM EDTA, 0.1 mM $Na_3VO_4$. Beads were then incubated with 30 μl KBZ reaction buffer (Echelon Bio-sciences, Utah, USA), and PI3K activity measured by incubating with $PIP_2$, and measuring $PIP_3$ pro-duction by a competitive ELISA as per the manufacturer's instructions (PI3-Kinase Activity ELISA: Pico, Echelon Biosciences).

## Statistical analysis

All quantitative data are reported as mean ± SEM of n observations. Statistical evaluation was per-formed using Student's t-test (where two variables were analysed) or one-way analysis of variance (ANOVA) (Prism 6 GraphPad Software, San Diego, CA) followed by Dunnett's or Bonferroni multiple comparison posthoc test, taking a probability $p < 0.05$ as statistically significant.

## Acknowledgements

This work was supported by British Heart Foundation grants (RG/10/15/28578, PG/10/6028496, RG/15/10/31485, BHF CRE RE/08/004/23915), Royal Society International exchange scheme grant (IE120747) and by US National Institutes of Health grants HL098055 and DK095684.

# Additional information

## Funding

| Funder | Grant reference number | Author |
| --- | --- | --- |
| British Heart Foundation | RG/10/15/28578 | Asif J Iqbal<br>Eileen McNeill<br>Keith M Channon<br>David R Greaves |
| BHF Centre of Research Excellence, Oxford | RE/08/004/23915 | Asif J Iqbal<br>David R Greaves |
| British Heart Foundation | PG/10/6028496 | Asif J Iqbal<br>Eileen McNeill<br>Keith M Channon<br>David R Greaves |
| British Heart Foundation | RG/15/10/31485 | Asif J Iqbal<br>Eileen McNeill<br>Keith M Channon<br>David R Greaves |
| National Institutes of Health | HL098055 | Tessa J Barrett<br>Edward A Fisher |
| National Institutes of Health | DK095684 | Tessa J Barrett<br>Edward A Fisher |
| Royal Society | IE120747 | David R Greaves<br>Edward A Fisher |

The funders had no role in study design, data collection and interpretation, or the decision to submit the work for publication.

## Author contributions

AJI, TJB, Conception and design, Acquisition of data, Analysis and interpretation of data, Drafting or revising the article; LT, EM, AM, CR, AC, MHB, Acquisition of data, Analysis and interpretation of data, Drafting or revising the article; GEW, Acquisition of data, Analysis and interpretation of data, Contributed unpublished essential data or reagents; DC, Conception and design, Acquisition of data, Analysis and interpretation of data; JAD, SLH, Provided reagents and technical assistance, Acquisition of data, Analysis and interpretation of data, Contributed unpublished essential data or reagents; MZ-D, Designed the purification strategy for the recombinant forms of apoA1 used, as well as prepared these materials; KMC, DRG, EAF, Conception and design, Analysis and interpretation of data, Drafting or revising the article

## Author ORCIDs

Asif J Iqbal, http://orcid.org/0000-0002-3224-3651

Lewis Taylor, http://orcid.org/0000-0003-4622-9890

Edward A Fisher, http://orcid.org/0000-0001-9802-143X

## Ethics

Human subjects: Human blood from anonymous healthy donors was obtained in the form of leukocyte cones from the NHS Blood and Transplant service. Leukocyte cones contain waste leukocytes isolated from individuals donating platelets via apharesis, and consist of a small volume (~10ml) of packed leukocytes with few red blood cells or platelets.

Animal experimentation: UK animal studies were conducted with ethical approval from the Dunn School of Pathology Local Ethical Review Committee and in accordance with the UK Home Office regulations (Guidance on the Operation of Animals, Scientific Procedures Act, 1986). All USA animal experiments were carried out according to the guidelines of the National Institutes of Health and approved by the New York University Institutional Animal Care and Use Committee (Protocol 102090).

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
