## [Decision Letter]

Thank you for submitting your article "Acute exposure to apolipoprotein A1 inhibits macrophage chemotaxis in vitro and monocyte recruitment in vivo" for consideration by *eLife*. Your article has been favorably evaluated by Tadatsugu Taniguchi (Senior editor) and three reviewers, one of whom is a member of our Board of Reviewing Editors. The following individual involved in review of your submission has agreed to reveal their identity: James Pease (peer reviewer).

The reviewers have discussed the reviews with one another and the Reviewing Editor has drafted this decision to help you prepare a revised submission.

Summary:

The manuscript from Iqbal et al., investigates acute effects of apoA1 on macrophage chemotaxis. They find that apoA1 inhibits macrophage and human monocyte chemotaxis to a range of chemokine and non-chemokine ligands. Exposure of mouse bone marrow monocytes to apoA1 in vitro reduces their ability to migrate into the peritoneal cavity in response to an inflammatory stimulus after adoptive transfer. They present evidence that the mechanism of inhibition of chemotaxis is through depletion of cholesterol from lipid rafts and reduced signaling through AKT. The findings are novel and interesting and may be of significance with respect to therapeutic applications of apoA1. Overall there was general enthusiasm for this manuscript for all three reviewers. One reviewer commented, 'The findings certainly provide a new venue for the utility of acute treatment apo-A1 and functionally similar agents in preventing the pro-inflammatory effects of monocytes/macrophages recruited to sites of inflammation.' Another reviewer commented, 'This is an interesting manuscript which I enjoyed reading and which I think the readership of the journal will certainly find of interest.'

The following major concerns were raised that would need to be satisfactorily addressed for the manuscript to be suitable for acceptance.

1) Specificity for apoA1. The experiments as currently presented lack a 'negative control'. Both wild type apoA1 and ox-apoA1 have the same activity. The findings would be strengthened by comparison with a mutant form of apoA1 that is unable to mediate cholesterol removal by either ABCA1-dependent or ABCA1-independent mechanisms.

2) The evidence that apoA1 blocks monocyte/macrophage chemotaxis by reducing lipid raft cholesterol remains correlative. Can the effect of apoA1 be blocked by cholesterol loading?

3) Following along the lines of points 1 and 2, the experiment shown in Figure 7 should be repeated to include testing effects of ox-apoA1, to confirm that it also depletes cholesterol from lipid raft fractions.

4) In the subsection “Decreased Akt cell signaling by apoA1 is associated with modulation of membrane lipid rafts”, Is it known how the comparative levels of circulating apoA1 in the Tg mouse compared to the WT? How does this compare with the concentrations of apoA1 that the isolated macrophages experience in vitro?

5) Figure 6 The competitive ELISA performed on RAW cells treated with C5a shows a trend for a reduction in PIP_3_ production following apoA1 treatment. If possible, it would be nice to see this repeated at least one more time to see if anything statistically significant can be drawn from it.

6) The second paragraph of the subsection “Acute exposure to apoA1 significantly reduces murine macrophage chemotactic responses” and the second paragraph of the Discussion claim that apoA1 treatment reduces the velocity and directionality of the migration, although the statistics in the table (Panel M) suggest these were not significantly different. It would be good to go back to the raw data in Figure 1 and analyze the ratio of track length to Euclidean distance to get the directionality parameter. Significant differences between vehicle and apoA1 treatment would support these claims. It may be that the experiment is slightly underpowered to detect subtle differences in these parameters, although there certainly appears to be a trend for their reduction with apoA1 treatment.

7) It would be excellent to support this aspect of the work with a similar ibidi study using freshly isolated human monocytes to see if the same migration parameters are affected to similar degrees.

[Editors' note: further revisions were requested prior to acceptance, as described below.]

Thank you for submitting your article "Acute exposure to apolipoprotein A1 inhibits macrophage chemotaxis in vitro and monocyte recruitment in vivo" for consideration by *eLife*. Your article has been favorably evaluated by Tadatsugu Taniguchi (Senior editor) and a member of our Board of Reviewing Editors.

The Reviewing Editor has drafted this decision to help you prepare a revised submission.

Summary:

The revised manuscript from Iqbal, et al., satisfactorily addresses most of the major concerns raised during initial review. The results of one experiment, however, seem to run counter to the major conclusion that apoA1 exerts effects on macrophage chemotaxis by reducing the concentration of cholesterol in lipid rafts.

Main point 2 of the initial review was:

The evidence that apoA1 blocks monocyte/macrophage chemotaxis by reducing lipid raft cholesterol remains correlative. Can the effect of apoA1 be blocked by cholesterol loading?

The authors respond:

To assess if cholesterol loading of cells blocks the effect of apoA1, biogel elicited mouse peritoneal macrophages were isolated from C57BL/6 mice and incubated with CyD:cholesterol complexes to mediate cell loading (Figure 10). Once loaded, cells were incubated with apoA1 and their migratory capacity to CCL5 assessed (Figure 10). Under these conditions we find that the inhibitory effect of apoA1 is evident even when cells are cholesterol-loaded.

This result is not discussed or included in the revised manuscript. On the face of it, loading cells with cholesterol should have had the opposite effect on macrophage chemotaxis than cholesterol depletion by cyclodextran or apoA1, and at least partially blunted the effect of apoA1. It is possible that apoA1 still reduces lipid raft cholesterol under these conditions, but the fact that there was no measurable impact of cholesterol loading on the effects of apoA1 does not allay the original concern that the effects of apoA1 and lipid raft cholesterol are correlative.

Essential revisions:

A compelling reason for why the cholesterol loading experiment is not informative needs to be provided or the experiments performed in Figure 7 need to be repeated under conditions of cholesterol loading to establish at apoA1 reduces cholesterol content in raft fractions even in cholesterol loaded cells.

---

## [Author Response]

The following major concerns were raised that would need to be satisfactorily addressed for the manuscript to be suitable for acceptance.

1) Specificity for apoA1. The experiments as currently presented lack a 'negative control'. Both wild type apoA1 and ox-apoA1 have the same activity. The findings would be strengthened by comparison with a mutant form of apoA1 that is unable to mediate cholesterol removal by either ABCA1-dependent or ABCA1-independent mechanisms.

Human apoA1 (243 amino acid residues) contains 22-amino acid repeats that form amphipathic α-helices. Even a single helix is competent to mediate cholesterol efflux from cells. L-4F is a peptide with one copy of an amphipathic helix that mimics the secondary structure of apoA1’s lipid-binding domain (Nagao et al., 2014). We found that treatment of biogel elicited macrophages with L-4F significantly impaired their migratory capacity to CCL2 (Figure 8). As an appropriate negative control, we tested the effect of a scrambled peptide that did not contain α-helices and found that macrophage migration to CCL2 was unaltered compared to vehicle-treated cells (Figure 8). This experiment demonstrates that deleting all of the helixes would be necessary to make a truly incompetent molecule. We next considered using a C-terminal deletion mutant. This domain is the most non-polar segment of the human apoA1 molecule. Apoa1 mutants that delete all or parts of the C-terminal have a diminished ability to efflux cholesterol via ABCA-1 dependent pathways, however their ability to promote cholesterol efflux via aqueous diffusion is minimally impacted (Nagao et al., 2014, Gillotte et al., 1999). As published, using a C-terminal deletion mutant (in which amino acids 185-243 are deleted) at a concentration similar to that used in our in vitro studies, total cholesterol efflux was ~85% that of wild type apoA1 because of the presence of multiple 22-amino acid amphipathic α-helices. Not surprisingly, then, incubating macrophages with this mutant also inhibited their migration to CCL5 (Figure 8). Note that this result supports our contention that ABCA1-mediated efflux is not required for the anti-migratory effects of apoA1, consistent with 80% of cholesterol efflux in unloaded cells being mediated by the aqueous diffusion pathway (Adorni et al., 2007).

Author response image 1.Peptides incorporating apoA1 amphipathic α-helices impair macrophage migration.Biogel elicited macrophages were and incubated with either (**A**) 50 µg/mL L-4F (Ac-DWFKAFYDKVAEKFKEAF-NH_2_), 50 µg/mL scrambled peptide (sc-4F) (Ac-DWFAKDYFKKAFVEEFAK-NH_2_) or (**B**) 40 µg/mL apoA1, 40 µg/mL C-terminal deleted apoA1 (Δ185-243) or PBS as vehicle control, before being added to the upper chamber (4×10^5^/well) of a CIM-16 plate and allowed to migrate for 7-8 h toward 10 nM CCL2 or CCL5. Migration was measured by max-min or slope analysis. Data are expressed as mean + SEM, n = 2 biological replicates with 3-4 technical replicates per condition. Statistical analysis was conducted by one-way ANOVA with Dunnett’s multiple comparison post-test. * p,0.05 relative to 10 nM chemokine alone.**DOI:**
http://dx.doi.org/10.7554/eLife.15190.015

Given that there is not an appropriate mutant form of apoA1 lipoprotein to address the reviewer’s concern, we therefore turned to using cyclodextrin analogs. Carboxy-cyclodextrin (carboxy-CyD) has a significantly reduced cholesterol efflux capacity when compared to methyl-β-cyclodextrin (CyD) (Figure 9). We find that in line with its diminished ability to promote cholesterol efflux, carboxy-CyD is unable to significantly impair the migratory capacity of bio-gel elicited mouse peritoneal macrophages, when compared to control treated cells (Figure 9), thereby further confirming the critical relationship of cholesterol efflux to the inhibition of myeloid cell migration.

Author response image 2.Macrophage cholesterol efflux impairs macrophage migration.(**A**) Biogel elicited macrophages were trace labelled with [^3^H]cholesterol and incubated with either 1 mM methyl-β-cyclodextrin (CyD) or 1 mM carboxy-cyclodextrin (carboxy-CyD) and cholesterol efflux assessed over 4 h. (**B**) Biogel elicited macrophages were incubated with either 1 mM methyl-β-cyclodextrin (CyD) or 1 mM carboxy-cyclodextrin (carboxy-CyD) or PBS as vehicle control, before being added to the upper chamber (4×10^5^/well) of a CIM-16 plate and allowed to migrate for 7-8 h toward 10 nM CCL5. Migration was measured by slope analysis. Data are expressed as mean + SEM, n = 2 biological replicates with 3-4 technical replicates per condition. Statistical analysis was conducted by one-way ANOVA with Dunnett’s multiple comparison post-test. * p,0.05 relative to 10 nM CCL5 alone.**DOI:**
http://dx.doi.org/10.7554/eLife.15190.016

2) The evidence that apoA1 blocks monocyte/macrophage chemotaxis by reducing lipid raft cholesterol remains correlative. Can the effect of apoA1 be blocked by cholesterol loading?

To assess if cholesterol loading of cells blocks the effect of apoA1, biogel elicited mouse peritoneal macrophages were isolated from C57BL/6 mice and incubated with CyD:cholesterol complexes to mediate cell loading (Figure 10). Once loaded, cells were incubated with apoA1 and their migratory capacity to CCL5 assessed (Figure 10). Under these conditions we find that the inhibitory effect of apoA1 is evident even when cells are cholesterol-loaded.

Author response image 3.ApoA1 impairs the migratory capacity of cholesterol-loaded cells.Biogel elicited macrophages were incubated with methy-β-cyclodextrin:cholesterol complexes (0.5 mg/mL) for 1 h to load cells, or PBS as vehicle control. Cholesterol quantification of cells confirmed that the macrophages were loaded (not shown). (**A**) Following incubation, cells were washed and then treated for 1 h with apoA1 (40 µg/ml) before being added to the upper chamber (4×10^5^/well) of a CIM-16 plate and allowed to migrate for 7-8 h toward 10 nM CCL5. Migration was measured by slope analysis (**B**). Data are expressed as mean + SEM, n = 3 biological replicates with 3-4 technical replicates per condition. Statistical analysis was conducted by one-way ANOVA with Dunnett’s multiple comparison post-test. * p,0.05 relative to PBS treated cells (for efflux studies) or 10 nM CCL5 alone (for migration studies).**DOI:**
http://dx.doi.org/10.7554/eLife.15190.017

3) Following along the lines of points 1 and 2, the experiment shown in Figure 7 should be repeated to include testing effects of ox-apoA1, to confirm that it also depletes cholesterol from lipid raft fractions.

We thank the reviewer for this comment. We repeated the experiments shown in Figure 7 with the addition of ox-rapoA1 and observed a significant depletion of cholesterol from lipid rafts fractions. These data have been added to Figure 7 panels I and J and noted in the revised text at the end of Results.

4) In the subsection “Decreased Akt cell signaling by apoA1 is associated with modulation of membrane lipid rafts”, Is it known how the comparative levels of circulating apoA1 in the Tg mouse compared to the WT? How does this compare with the concentrations of apoA1 that the isolated macrophages experience in vitro?

For our in vitro studies we chose to use 40 µg/mL of apoA1, as at this concentration the efflux capacity of apoA1 has reached its saturation point, e.g. (Hoang et al., 2012). In apoA1 Tg mice, total apoA1 (murine plus human apoA1) is increased two fold compared to WT controls, with the concentration of apoA1 in Tg mice being approximately 252 mg/dL (2520 µg/mL) (Rubin et al., 1991, Chajek-Shaul et al., 1991). The average concentration ratio of interstitial fluid to plasma for apoA1 is between 0.07 to 0.12 (Sloop et al., 1987). Taking into account the interstitial fluid/plasma ratio, the effective apoA1 concentration a typical cell would see in vivois approximately 176-302 µg/mL. Even in WT mice, the apoA1 plasma concentration is approximately 110 mg/dL (Rubin et al., 1991), with a typical interstitial concentration estimated to be 77-132 µg/mL. These calculations indicate that our in vitro experiments employ an apoA1 concentration well within the physiological range of that achieved not only in the transgenic, but also the WT mice, and, thus, the results are not a consequence of a pharmacologic effect of apoA1.

5) Figure 6 The competitive ELISA performed on RAW cells treated with C5a shows a trend for a reduction in PIP_3_ production following apoA1 treatment. If possible, it would be nice to see this repeated at least one more time to see if anything statistically significant can be drawn from it

This has been repeated and data added to Figure 6= 4 independent experiments). With the additional data, the previous trend in reduction in PI3K activity following apoA1 treatment that we observed is now statistically significant.

6) The second paragraph of the subsection “Acute exposure to apoA1 significantly reduces murine macrophage chemotactic responses” and the second paragraph of the Discussion claim that apoA1 treatment reduces the velocity and directionality of the migration, although the statistics in the table (Panel M) suggest these were not significantly different. It would be good to go back to the raw data in Figure 1 and analyze the ratio of track length to Euclidean distance to get the directionality parameter. Significant differences between vehicle and apoA1 treatment would support these claims. It may be that the experiment is slightly underpowered to detect subtle differences in these parameters, although there certainly appears to be a trend for their reduction with apoA1 treatment.

We have gone back and reanalysed the data to calculate directionality, however this parameter was not different between vehicle and apoA1 treated cells. We still believe that apoA1 inhibits chemotaxis, as FMI was significantly reduced and migration distances were reduced, but we accept that in order to detect subtle changes in directionality this experiment may be underpowered.

However, we conducted new ibidi experiments with human monocytes, and the effect of apoA1 treatment was very striking (Figure 3). All parameters were dramatically reduced, including the new directionality parameter. Therefore, we believe that using these data we can substantiate the claim that apoA1 reduces the velocity and directionality of chemotaxis. We have updated the text to reflect this new analysis of the ibidi data (subsection “Acute exposure to apoA1 significantly reduces human monocyte chemotaxis”, first paragraph).

7) It would be excellent to support this aspect of the work with a similar ibidi study using freshly isolated human monocytes to see if the same migration parameters are affected to similar degrees.

We have addressed this point by including a new ibidi real time chemotaxis study on human monocytes at the end of Figure 3 (panels J-M). To further extend upon this comment we also performed flow chamber studies with human peripheral blood mononuclear cells (PBMCs) (Figure 3—figure supplement 1; discussed in subsection “Acute exposure to apoA1 significantly reduces human monocyte chemotaxis”, second paragraph) to highlight that acute treatment with apoA1 also inhibits PBMC rolling, adhesion, and transmigration under flow and that these effects were independent of changes in receptor and adhesion molecule expression.

[Editors' note: further revisions were requested prior to acceptance, as described below.]

Summary:

The revised manuscript from Iqbal, et al., satisfactorily addresses most of the major concerns raised during initial review. The results of one experiment, however, seem to run counter to the major conclusion that apoA1 exerts effects on macrophage chemotaxis by reducing the concentration of cholesterol in lipid rafts.

Main point 2 of the initial review was:

The evidence that apoA1 blocks monocyte/macrophage chemotaxis by reducing lipid raft cholesterol remains correlative. Can the effect of apoA1 be blocked by cholesterol loading?

The authors respond:

To assess if cholesterol loading of cells blocks the effect of apoA1, biogel elicited mouse peritoneal macrophages were isolated from C57BL/6 mice and incubated with CyD:cholesterol complexes to mediate cell loading (Figure 10). Once loaded, cells were incubated with apoA1 and their migratory capacity to CCL5 assessed (Figure 10). Under these conditions we find that the inhibitory effect of apoA1 is evident even when cells are cholesterol-loaded.

This result is not discussed or included in the revised manuscript. On the face of it, loading cells with cholesterol should have had the opposite effect on macrophage chemotaxis than cholesterol depletion by cyclodextran or apoA1, and at least partially blunted the effect of apoA1. It is possible that apoA1 still reduces lipid raft cholesterol under these conditions, but the fact that there was no measurable impact of cholesterol loading on the effects of apoA1 does not allay the original concern that the effects of apoA1 and lipid raft cholesterol are correlative.

Essential revisions:

A compelling reason for why the cholesterol loading experiment is not informative needs to be provided or the experiments performed in Figure 7 need to be repeated under conditions of cholesterol loading to establish at apoA1 reduces cholesterol content in raft fractions even in cholesterol loaded cells.

As suggested by the reviewer, new experiments were performed using the human macrophage cell line, THP-1. The data (Figure 1, below) effectively demonstrate that apoA1 can significantly remove cholesterol from membrane lipid rafts of cholesterol loaded cells. We have now incorporated these new findings along with the findings from our previous resubmission into the text of manuscript (below) and in new supplementary Figure 7—figure supplement 1.

Author response image 4.ApoA1 depletes membrane lipid rafts of cholesterol in cholesterol-loaded THP-1 cells.Human THP-1 macrophages were incubated with methyl-β-cyclodextrin:cholesterol complexes (0.5 mg/mL) for 1 h to cholesterol-load cells prior to incubation for 1 h with 40 µg/mL apoA1 or PBS. Cells were then homogenized in a 0.2% Triton-X 100 containing buffer prior to 5-45% sucrose density gradient centrifugation. Fractions were collected from the top (fraction 1, lightest fraction) to the bottom (fraction 10, heaviest fraction), and cholesterol content of each fraction quantified. The sums of the cholesterol content in lipid raft fractions (fractions 2-4), and non-raft fractions (fractions 8-10), are shown in panels (**A**) and (**B**) respectively. Statistical analysis was conducted by using Student’s t-test, taking *P* < 0.05 as statistically significant.**DOI:**
http://dx.doi.org/10.7554/eLife.15190.018

The updated text now appears in the Discussion:

“In addition, the inhibitory effect of apoA1 on macrophage chemotaxis to CCL5 is not affected by cholesterol loading of the cell (data not shown), consistent with apoA1’s ability to promote cholesterol removal from membrane lipid rafts of these cells (Figure 7—figure supplement 1).”